



# Concurrent aerodynamic design of the wing and the turbines of airborne wind energy systems

Filippo Trevisi[1], Gianni Cassoni[2,3], Mac Gaunaa[4], and Lorenzo Fagiano[1]

[1]Department of Electronics, Information and Bioengineering, Politecnico di Milano, Via Ponzio, 34, 20156 Milano, Italy
[2]Department of Aeronautical Technologies Engineering, Roma Tre University, Via della Vasca Navale, 79, 00146 Rome, Italy
[3]Department of Aerospace Science and Technology, Politecnico di Milano, Via La Masa, 34, 20156 Milano, Italy
[4]DTU Wind Energy, Frederiksborgvej 399, 4000 Roskilde, Denmark

**Correspondence:** F. Trevisi (filippo.trevisi@polimi.it)

**Abstract.** The aerodynamic design of the aircraft of fly-gen Airborne Wind Energy Systems, named windplane here, is one of the main aspects determining their power production, but it is still a largely unexplored problem. To this end, an engineering model for the aerodynamics of the onboard turbines, the aerodynamics of the wing and their interactional aerodynamics is developed and coupled to a steady-state windplane model and a far-wake model. This comprehensive model is then used to design the windplane aerodynamics for a given wingspan. Initially, a design space exploration study reveals that placing the turbines at the wing tips and rotating them inboard down increases the power production compared to other locations and rotation direction. This is because the turbines' wake swirl reduces the wing induced drag, especially when they are placed at the wing tips. Moreover, conventional efficient airfoils are found to be optimal for windplanes. Later, NACA4421 airfoils are used for the design of the wing and the turbines, placed at the wing tips. The optimal trapezoidal wing, modeled with constant twist, has an aspect ratio of 5.1, a taper ratio of 0.60 and the onboard turbines operate at a design low tip speed ratio of 1.9 to increase the wake swirl. The results from the vortex models of the wing, the turbines, and their interaction is extensively compared with the lifting line, the vortex lattice method and the vortex particle method implemented in the well-established code DUST, finding very good agreement. Finally, the windplane is studied with DUST at different wing angles of attack and at different turbine tip speed ratios to characterize its behavior away from the design point.

## 1 Introduction

Airborne Wind Energy Systems (AWESs) harvest wind power by means of a fully autonomous aircraft connected to the ground with a tether. AWESs can be classified in crosswind, tether aligned and rotational (Vermillion et al. (2021)). Most of industrial and research activities focus on crosswind AWES, whose working principles were first described by Loyd (1980). Crosswind AWESs move in a fast motion roughly perpendicular to the wind direction and generate power with onboard small wind turbines (fly-gen AWES), as studied in this paper, or by pulling the tether and unwinding a generator on the ground (pumping AWES). For more details, Fagiano et al. (2022) give an overview of the state-of-the-art of these systems.

Fly-gen AWESs fly figure-of-eight or circular trajectories, generating power with the onboard turbines and transmitting it to the ground via the tether, which has a structural and an electrical component. The aircraft of fly-gen AWES is termed windplane



in this paper. It is a rather unique system, whose design is far from being fully understood. This is due to the need to operate in
very different phases (take-off and landing, power production, transitions), in a broad range of wind speeds, and to maximize the
converted power while meeting structural and controllability requirements. Currently, fly-gen systems are actively developed
by companies such as Kitekraft GmbH (2025), WindLift (2025) and were developed by Makani Technologies LLC (2025),
which stopped operations in 2020.

KiteKraft's design is based on a box wing, with turbines distributed along the span of both wings. The AWES flies figure-
of-eight trajectories and operates the windplane, characterized by multi-elements airfoils, at very high lift coefficients (Bauer
et al. (2018)). The onboard turbines design is approached with the methodologies developed by Frirdich (2019) for the power
generation and to sustain the AWES during hover. The wing and the onboard turbine designs are performed separately. WindLift
design is not disclosed and only a limited amount of information is publicly available. Their design is characterized by a single
wing with four turbines mounted on pylons. The Makani development, on the other hand, is well documented in sources
available in the public domain. After the shut-down, Makani engineers released two reports summarizing the internal design,
analyses and test methodologies (Echeverri et al. (2020a) and Echeverri et al. (2020b)). Makani's last prototype M600 had a
wingspan of 26 m, an aspect ratio of 20 and had 8 turbines mounted on pylons along the wing. Before the shut-down Makani
engineers re-designed the AWES to overcome major shortcomings, resulting in the MX2 design. The MX2 was designed to
feature a wingspan of 26 m, an aspect ratio of 12.5 and 8 turbines. The wing's and turbines' design were carried out separately.
The turbines are designed to generate power and to provide the thrust to sustain the AWES during hover. Their wing is designed
to maximize the power harvesting factor[1], defined by Diehl (2013), as explained by Tucker (2020) and by an earlier paper of
Vander Lind (2013). Wings designed to maximize the power harvesting factor have a high aspect ratio (similar to gliders) and
use unconventional airfoils, designed to maximize the metric $C_l^3/C_d^2$.

Recently, Trevisi (2024) introduced a new design methodology for windplanes, performing the aerodynamic design per a
given wingspan. This is achieved by maximizing a newly-defined power coefficient [2] (Trevisi et al. (2023a)), and it results in an
optimal wing with low aspect ratio (approximately between 4 and 7) and conventional efficient airfoils. This new methodology
is improved in this paper to concurrently design the main wing and the onboard turbines. Three turbine positions are considered:
in front, at the wing tips and above or below the wing. If they are placed above or below, they need pylons, which have an
associated aerodynamic drag and decrease the system performances. If they are placed in front of the wing, they reduce the
apparent wind speed felt by the involved wing aerodynamic sections, which in principle would lead to a lower lift force. On the
contrary, airplanes with propellers in front of the wing achieve an increase in aerodynamic lift. In fact, another aerodynamic
effect generated by the onboard rotors is the swirl in their wake. Airplane designers make use of this effect by placing the
onboard propellers at the wing tip (Snyder and Zumwalt (1969); J. Patterson and Bartlett (1985)), such that the aerodynamic
sections of the wing after the rotor would experience a beneficial change in inflow angle, leading to a decrease in induced
drag. Miranda and Brennan (1986) started the theoretical and numerical investigation of the decrease in induced drag due to tip
propellers and to tip turbines. They developed a vortex model, similar to the one presented in this paper, to assess the benefit

---

[1]This power harvesting factor uses the wing planform area as the reference area

[2]This power coefficient uses a disk with radius equal to the wingspan as the reference area





of placing the rotors at the tip. Recently, Sinnige et al. (2019b) performed wind tunnel tests of this configuration comparing it with conventional layout and studied the performance of tip propellers in energy-harvesting conditions (Sinnige et al. (2019a)). Taking inspiration from this aeronautical experience, we here design the windplane onboard turbines with engineering models.

We design the rotors for the generation phase only, and do not consider whether they generate enough thrust in the hover state. This is done to focus on the ideal aerodynamic design of the windplane to fulfill its main functionality, i.e. power generation. The design can be later modified to cope with the take-off and landing phases, which is less demanding in terms of power ratings (Fagiano and Schnez (2017)). We validate the proposed framework with the lifting line, the vortex lattice method and the vortex particle method implemented in DUST (Tugnoli et al. (2021)). This code is open-source and has been validated

for wing-propeller studies (Niro et al. (2024)). Finally we use DUST to characterize the wing and rotor design away from the design point. The vortex particle method has recently been used by Mehr et al. (2024) to analyze the Makani prototype (characterized by rotors mounted on pylons) and investigate the effects of rotor rotation direction and vertical and streamwise rotor position on the overall aerodynamic performance, without however moving the rotors in front of the wing and along the wing span.

This paper is organized as follows: In Sect. 2, the aerodynamic modeling framework is introduced; in Sect 3, the aerodynamic optimization problem formulation is introduced and a design space exploration study on the turbine positions and on the optimal airfoils characteristics is carried out; in Sect. 4, the vortex particle code DUST, used for validation and analysis, is introduced; in Sect. 5, the optimal design problem is solved, the solution is compared with DUST and the design is studied away from the design point; In Sect. 6, the main conclusions are discussed and future work suggested. A list of symbols is given at the end of

the paper.

## 2 Windplane aerodynamics model

We start by formulating the modeling framework for the optimal design problem of the wing and the rotors. The model is meant to be as simple as possible, while still capturing the relevant physical phenomena.

### 2.1 Windplane steady-state model

We assume a steady-state equilibrium, neglecting gravity and assuming a circular fully crosswind trajectory with steady wind. Indeed, Trevisi (2024) shows that the aerodynamic problem is not influenced by the dynamic problem. Referring to Fig. 1, the force balance is written in the cylindrical coordinate system $(\mathbf{e}_\tau, \mathbf{e}_r, \mathbf{e}_z)$, where $\mathbf{e}_\tau$ points tangentially, $\mathbf{e}_r$ radially and $\mathbf{e}_z$ axially.



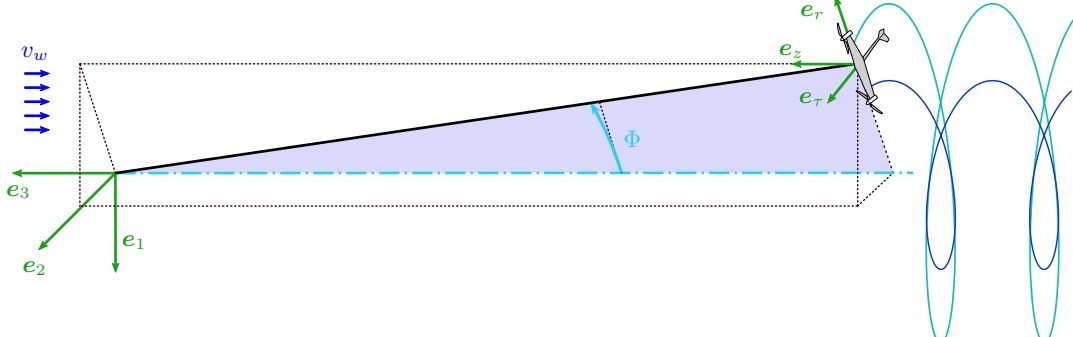

**Figure 1.** Ground coordinate system $(\mathbf{e}_1, \mathbf{e}_2, \mathbf{e}_3)$ and cylindrical coordinate system $(\mathbf{e}_\tau, \mathbf{e}_r, \mathbf{e}_z)$.

The aerodynamic forces and the main wake components are shown in Fig. 2. Considering the velocity triangle and the inflow angle in the near wake $\gamma_n$, the force balances in the three axes are

$$
\begin{cases}
L\sin(\gamma_n) - (D_p + D_i + T_t)\cos(\gamma_n) = 0 \\
m\dfrac{u^2}{R_0} = T_z \tan(\Phi) \\
T_z = L\cos(\gamma_n) + (D_p + D_i + T_t)\sin(\gamma_n)
\end{cases}
\tag{1}
$$

where $L$ is the aerodynamic lift, $D_p$ the parasite drag, $D_i$ the induced drag, $T_t$ the turbine thrust, $m$ the system mass (equal to the aircraft mass plus one third of the tether mass, as derived by Trevisi et al. (2020)), $u$ the tangential velocity, $T_z$ the axial component of the tether force and $\Phi$ the opening angle of the circular trajectory.

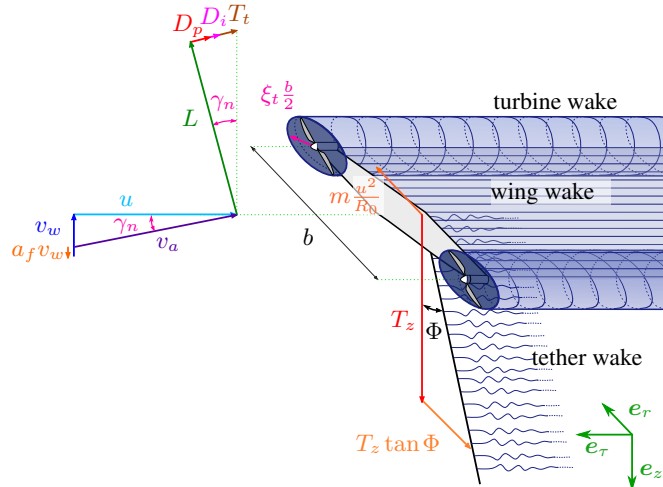

**Figure 2.** Velocity triangle, forces acting on the windplane in crosswind steady state and main wake components.





The aerodynamic lift $L$, defined to be perpendicular to the apparent velocity $v_a$, is

$$L = \frac{1}{2}\rho A C_L v_a^2, \tag{2}$$

where $\rho$ is the air density, $C_L$ is the wing lift coefficient, $v_a$ is the apparent wind speed and the wing area $A$ is

$$A = \frac{b^2}{AR}, \tag{3}$$

where $b$ is the wing span and $AR$ is the wing aspect ratio.

The aerodynamic parasite drag is

$$D_p = \frac{1}{2}\rho A C_{D,p} v_a^2, \tag{4}$$

where $C_{D,p}$ is the parasite drag coefficient, defined as

$$C_{D,p} = C_{D,a} + C_{D,te}. \tag{5}$$

$C_{D,a}$ is the three-dimensional coefficient due to the airfoils and $C_{D,te}$ is the equivalent tether drag (Trevisi et al. (2020))

$$C_{D,te} = C_{d,te}\frac{L_{te}D_{te}}{4A}, \tag{6}$$

where $C_{d,te}$ is the tether section drag coefficient, $L_{te}$ the tether length and $D_{te}$ the tether diameter.

The aerodynamic induced drag is

$$D_i = \frac{1}{2}\rho A C_{D,i} v_a^2, \tag{7}$$

where $C_{D,i}$ is the induced drag coefficient of the near wake. Indeed, Trevisi et al. (2023b) show that the induced velocities due to the near wake (first half rotation of the helicoidal wake of Fig. 5) can be taken into account with an induced drag coefficient computed assuming the trailed vortex filaments as straight, as done in this work. More details about this modeling approach are given in Sect. 2.4.

The thrust force produced by the onboard wind turbines is

$$T_t = \frac{1}{2}\rho A_t C_{T,t} v_a^2, \tag{8}$$

where $C_{T,t}$ is the thrust coefficient of the onboard wind turbines and $A_t$ is the total rotor area. Defining the onboard wind turbine radius as function of the wing semi-span as

$$R_t = \xi_t \frac{b}{2}, \tag{9}$$

the total rotor area $A_t$, assuming two turbines, is

$$A_t = 2 \cdot \pi R_t^2 = \frac{\pi \xi_t^2}{2}b^2. \tag{10}$$





The employed model to estimate the thrust coefficient $C_{T,t}$ of the onboard turbines is described in Sect. 2.2, while the models for the aerodynamic coefficients $C_{D,a}$ and $C_{D,i}$ are detailed in Sect. 2.3.

The force balance along the tangential direction (first equation in (1)), can be written as

$$\tan(\gamma_n) = \frac{D_p + D_i + T_t}{L} = \frac{1}{E}. \tag{11}$$

where $E$ is the windplane aerodynamic efficiency, including the thrust of the turbines.

The inflow angle in the near wake, $\gamma_n$, can be found by considering the tangential velocity to the plane's path $u$ and by subtracting the induced velocity due to the far wake $a_f v_w$ to the free wind velocity $v_w$ (Trevisi et al. (2023b))

$$\tan(\gamma_n) = \frac{v_w(1 - a_f)}{u} = \frac{(1 - a_f)}{\lambda}, \tag{12}$$

where $\lambda = \frac{u}{v_w}$ is the wing speed ratio.

The wing speed ratio $\lambda$ can be found by equating Eq. (11) and Eq. (12) and using the definition of the aerodynamic forces

$$\lambda = E(1 - a_f) = \frac{\frac{1}{2}\rho A C_L v_a^2 (1 - a_f)}{\frac{1}{2}\rho A (C_{D,p} + C_{D,i} + \frac{A_t}{A} C_{T,t}) v_a^2} = \frac{C_L(1 - a_f)}{C_{D,p} + C_{D,i} + \frac{\pi}{2} AR \xi_t^2 C_{T,t}}. \tag{13}$$

The power equation can be written with respect to the onboard turbines as

$$P = \frac{1}{2}\rho A_t C_{P,t} v_a^3 \approx \frac{1}{2}\rho A_t C_{P,t} \lambda^3 v_w^3 = \frac{1}{2}\rho A_t C_{P,t} E^3 (1 - a_f)^3 v_w^3, \tag{14}$$

where $C_{P,t}$ is the power coefficient of the onboard wind turbines and large values of $\lambda$ are assumed ($\lambda^2 \gg 1$). In practice, the wing speed ratio range from $\lambda \approx 5$ to $\lambda \approx 10$, thus justifying this assumption.

The windplane power coefficient, using as reference area the disk with radius the wingspan (Trevisi et al. (2023a)), is

$$C_P = \frac{P}{\frac{1}{2}\rho \pi b^2 v_w^3} = C_{P,t} \frac{\xi_t^2}{2} \lambda^3 = C_{P,t} \frac{\xi_t^2}{2} E^3 (1 - a_f)^3. \tag{15}$$

The windplane thrust coefficient, similar to the conventional turbines' thrust coefficient, is defined to inform about the aerodynamic force applied to the wind as

$$C_T = \frac{T_z}{\frac{1}{2}\rho \pi b^2 v_w^2} \approx \frac{C_L}{\pi AR} \lambda^2 = \frac{C_L}{\pi AR} E^2 (1 - a_f)^2, \tag{16}$$

where small inflow angles in the near wake, i.e. $\gamma_n \ll 1$, are assumed, such that the axial component of the tether force can be approximated with the lift force $T_z \approx L$ (third equation in (1)). Writing the turning radius as $R_0 = L_{te} \sin\Phi$, the radial equilibrium (second equation in (1)) can be formulated to find the opening angle $\Phi$ such that the wingspan is fully crosswind (i.e. the lift is not used for turning)

$$\sin\Phi \tan\Phi = \frac{m}{\frac{1}{2}\rho A C_L L_{te}}. \tag{17}$$





## 2.2 Onboard turbines model

In order to evaluate the onboard turbines performance and the effect of their wake on the wing, we use the vortex cylinder model introduced by Branlard and Gaunaa (2016), which accounts for the radially varying circulation by superposing the vortex cylinders models developed by Branlard and Gaunaa (2015). The model is an actuator disk model like the momentum-based framework that BEM models are based on. The added value provided by the vortex cylinder model is that it includes the effect of the pressure drop due to wake rotation that is neglected in the momentum-based models. The importance of this effect increases as the tip speed ratio is decreased.

Let us define the parameter $k$ as $k(r_t) \equiv \frac{\Omega_t \Gamma_t(r_t)}{\pi u^2}$, where $\Gamma_t(r_t)$ is the bound circulation of the annulus, $\Omega_t$ the rotor angular velocity, $u$ the inflow velocity to the rotor and $r_t \in [0, R_t]$ the radial coordinate. Then, the rotor quantities can be evaluated for a given radial distribution of the parameter $k(r_t)$ and a given tip speed ratio $\lambda_t$. The tangential induction at the turbine radius $r_q$ can be evaluated for a given value of $k_q(r_q)$ as

$$a'_{t,q} = \frac{k_q}{4\lambda^2_{r_q,t}} \tag{18}$$

where the local speed ratio is $\lambda_{r_q,t} = \lambda_t \frac{r_q}{R_t}$. The dimensional tangential velocity (swirl) at the rotor disk can be then evaluated as

$$w_{t,q} = a'_{t,q}\lambda_{r_q,t} u. \tag{19}$$

The local thrust coefficient is defined as

$$C_{t,q} \equiv k_q\left(1 + \frac{k_q}{4\lambda^2_{r_q,t}}\right), \tag{20}$$

such that the rotor thrust coefficient is

$$C_{T,t} = \frac{1}{\pi R_t^2}\sum_q C_{t,q}\pi\left(r^2_{q+1} - r^2_q\right). \tag{21}$$

The effect of the rotation of the wake is a decrease of the pressure toward the rotational axis. The part of the local thrust coefficient corresponding to this is the rotational thrust coefficient, which is

$$C_{t,\mathrm{rot},q} \equiv \sum_{j>q}\left(\frac{k_j}{2}\right)^2\left[\frac{1}{\lambda^2_{r_{j-1},t}} - \frac{1}{\lambda^2_{r_j,t}}\right]. \tag{22}$$

The induction at each radial station is

$$a_{t,q} = \frac{1}{2}\left(1 - \sqrt{1 - C_{t,q} + C_{t,\mathrm{rot},q}}\right), \tag{23}$$

such that the reduction in axial velocity at the rotor disk is

$$u_{t,q} = a_{t,q}u. \tag{24}$$





The local power coefficient is

$$C_{p,q} = k_q \left(1 - a_q\right), \tag{25}$$

and the rotor power coefficient is

$$C_{P,t} = \frac{1}{\pi R_t^2} \sum_q C_{p,q} \pi \left(r_{q+1}^2 - r_q^2\right). \tag{26}$$

This model enables the estimation of the turbine thrust coefficient $C_{T,t}$ (Eq. 21) and power coefficient $C_{P,t}$ (Eq. 26) for a given distribution of the parameter $k(r_t)$ and tip speed ratio $\lambda_t$. Moreover, it provides the velocities induced in the turbine wake[3], which are twice the velocities in the rotor disk. Veldhuis (2004) highlights that a swirl recovery factor of $1/2$ should be included to properly model the rotor-wing interaction, when the velocities in the rotor wake are considered as inputs to the wing. The reduction in the slipstream induction and swirl can be attributed to the wing induced upwash. The correct induced

velocities to be used as inputs for the wing are then the induced velocities in the rotor disk. This is explained by Miranda and Brennan (1986) as a generalization of Munk stagger theorem (Prandtl (1924)), which is used to compute the induced drag of multiple wings (e.g. main wing and horizontal stabilizer). This theorem states that the total induced drag of systems of multiple wings with a fixed Trefftz-projected bound circulation is independent of the streamwise location of each of the wings. This shows that the total induced drag of a multiple-wing system can be calculated by moving the wings to the same streamwise

position. Similarly, the correct induced drag of the rotor-wing system in the present case is calculated by moving the rotors to the same streamwise position of the wing. For this reason, the induced velocities $w_t$ and $u_t$ (Eqs. 19 and 24), which are the values at the rotor disk (equal to half the values in the turbine far wake), are used as inputs for the wing.

These considerations also point out that placing the turbines after the wing would not change the overall induced drag, as long as they are placed in the same stream-wise location. Thus, in this paper, we only study the case in which the rotors are

placed in front of the wing.

## 2.3 Wing model

We introduce here the wing model, based on Weissinger (1947) lifting line theory. An infinitesimal segment of vortex filament $dl$, located at $l$ and with vortex strength $\Gamma$, induces a velocity $dV$ on an arbitrary point $P$. Defining the distance from $dl$ to $P$ as $r$, the induced velocity $dV$ can be evaluated with Biot-Savat law

$$dV = \frac{\Gamma}{4\pi} \frac{dl \times r}{|r|^3}, \tag{27}$$

Referring to Figure 3, we model the wing as a straight lifting line pointing along the $y$ direction, discretized in $N_p$ elements. The chord direction of the airfoil at the wing center is taken to be equal to the $x$ direction. Each element $p$ is characterized by a bound vortex of strength $\Gamma_p$ located along the direction of the quarter-chord line. According to the Helmholtz's vortex

---

[3]The tangential velocity jumps from zero just in front of the disk to twice the disk value after the disk. The axial induced velocity undergoes a smooth variation from zero far upstream to twice the disk induction in the far wake behind the disk. In the regions close to the disk, the axial inductions depend on the loading of all annular strips.





theorem, a vortex must extend to the boundaries of the fluid, so that each element trails two vortexes of strength equal to

$\Gamma_p$ from its extremes. The number of trailed vortices is then $N_p + 1$, with intensity $\gamma_p = \Gamma_p - \Gamma_{p-1}$. According to Pistolesi'

theorem, following the implementation by Damiani et al. (2019), we evaluate the induced velocities at the chord-wise position

of $\frac{3}{4}$ chord of each element. The distance $\boldsymbol{r}$ of the evaluation point $\boldsymbol{P}_j = [-\frac{c_j}{2}; y_j; 0]$, with respect to the reference system

placed on the quarter-chord line, from a generic point located at $\boldsymbol{l} = [x; y; 0]$ is

$$
\boldsymbol{r} = \begin{bmatrix} -\frac{c_j}{2} - x \\ y_j - y \\ 0 \end{bmatrix} = r_j \begin{bmatrix} -\sin(\theta_j) \\ \cos(\theta_j) \\ 0 \end{bmatrix},
\tag{28}
$$

with the angle $\theta_j$ being

$$
\tan(\theta_j) = \frac{\frac{c_j}{2} + x}{y_j - y}.
\tag{29}
$$

The velocity induced by an infinitesimal portion of the bound vortex located at $\boldsymbol{l} = [0; y; 0]$ on the evaluation point $\boldsymbol{P}_j$ can

be evaluated with Eq. (27), where $d\boldsymbol{l} = [0; dy; 0]$ and $\boldsymbol{r}$ is given in Eq. (28),

$$
dw_j|_{bound} = \frac{\Gamma}{4\pi} \frac{\sin(\theta_j)}{r_j{}^2} dy = \frac{\Gamma}{4\pi} \frac{\sin(\theta_j)}{c_j/2} d\theta_j,
\tag{30}
$$

with $dy = \frac{1}{c_j/2} \frac{(y_j - y)^2}{\cos(\theta_j)^2} d\theta_j$, obtained differentiating Eq. (29) with $x = 0$.

The velocity induced by the finite bound vortex $p$ is then

$$
\Delta w_{p,j}|_{bound} = \frac{\Gamma_p}{4\pi} \frac{1}{c_j/2} \int_{\theta_{p,j}}^{\theta_{p+1,j}} \sin(\theta_j) d\theta_j = -\frac{\Gamma_p}{4\pi} \frac{\cos(\theta_{p+1,j}) - \cos(\theta_{p,j})}{c_j/2}.
\tag{31}
$$

The velocity induced by the trailed vortex $p$ on the evaluation point $\boldsymbol{P}_j$ is evaluated by integrating Eq. (27), where $d\boldsymbol{l} = [dx; 0; 0]$ and $\boldsymbol{r}$ is given in Eq. (28),

$$
\Delta w_{p,j}|_{trailed} = \frac{\gamma_p}{4\pi} \int_{-\infty}^{0} \frac{y_j - y_p}{r_j{}^3} dx = \frac{\gamma_p}{4\pi} \frac{1}{y_j - y_p} \int_{-\pi/2}^{\theta_{p,j}} \cos\theta_j d\theta_j = \frac{\gamma_p}{4\pi} \frac{1 + \sin\theta_{p,j}}{y_j - y_p},
\tag{32}
$$

with $dx = \frac{(y_j - y_p)}{\cos(\theta_j)^2} d\theta_j$, obtained differentiating Eq. (29) with $y = y_p$.

Since the induced velocities from the three dimensional vorticity distribution are evaluated at the $3c/4$, also the induced

velocity from the two-dimensional vorticity distribution need to be evaluated at the same point (Damiani et al. (2019)). The 2D

bound vortex induces a velocity at the evaluation point

$$
\Delta w_j|_{2D} = \frac{\Gamma_j}{2\pi} \frac{1}{c_j/2}.
\tag{33}
$$

The total induced velocity at the evaluation point $\boldsymbol{P}_j$ is then

$$
w_j = \sum_{p=1}^{N_p} (\Delta w_{p,j}|_{bound}) + \sum_{p=1}^{N_p+1} (\Delta w_{p,j}|_{trailed}) - \Delta w_j|_{2D}
\tag{34}
$$



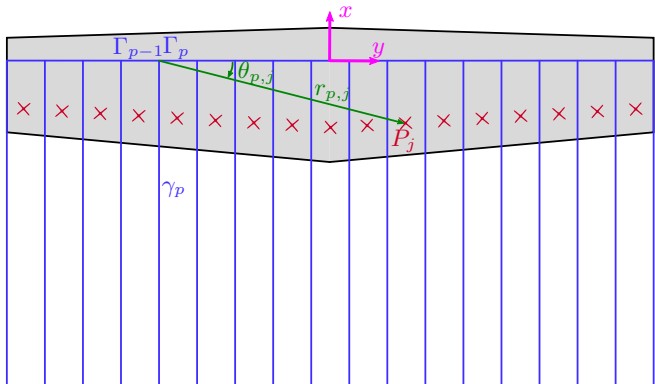

**Figure 3.** Wing geometry. The bound vorticity $\Gamma_p$ of the element $p$ and the vorticity $\gamma_p = \Gamma_p - \Gamma_{p-1}$ of the trailed vortex $p$ are shown in blue. The red crosses are the evaluation points, with $\boldsymbol{P}_j$ being the evaluation point of element $j$. The distance $r_{p,j}$ and the angular position $\theta_{p,j}$ of evaluation point $P_j$ with respect to the start of the vortex filament $p$ are shown in green.

The relative wind velocity at $3c/4$ of the aerodynamic section $j$ takes into account the free-stream velocity $\boldsymbol{v}_a$, the velocities induced by the turbines by the turbines $\boldsymbol{v}_{t,j}$ and by trailed vorticity $w_j$

$$\boldsymbol{v}_{r,j} = \overbrace{-v_a \begin{bmatrix} \cos(\alpha) \\ 0 \\ \sin(\alpha) \end{bmatrix}}^{\boldsymbol{v}_{\infty,j}}_{\boldsymbol{v}_a} + \underbrace{\begin{bmatrix} u_{t,j} \\ 0 \\ w_{t,j} \end{bmatrix}}_{\boldsymbol{v}_{t,j}} + \begin{bmatrix} 0 \\ 0 \\ w_j \end{bmatrix} \tag{35}$$

where $\alpha$ is the undisturbed wing angle of attack, defined as the angle between the chord line of the airfoil at the wing center ($x$ direction) and the undisturbed inflow.

The angle of attack used to compute the airfoil lift $C_l$ and drag coefficients $C_d$ is

$$\alpha_{j,3c/4} = \arctan\left(\frac{\boldsymbol{v}_{r,j}(3)}{\boldsymbol{v}_{r,j}(1)}\right) + \beta_j. \tag{36}$$

where $\beta_j$ is the local twist angle (i.e. the angle between the $x$ direction and the local airfoil chord line).

The lift $C_l$ and drag coefficient $C_d$ should however be projected according to the velocity triangle at the $c/4$ lines (Li et al. (2022)). The velocity at the quarter-chord line $\alpha_{c/4}$ ($\boldsymbol{P}_j = [0; y_j; 0]$) takes into account the free-stream velocity $\boldsymbol{v}_a$, the velocities induced by the turbines $\boldsymbol{v}_{t,j}$ and by trailed vorticity $w_{j,c/4}$

$$\boldsymbol{v}_{r,j,c/4} = \overbrace{-v_a \begin{bmatrix} \cos(\alpha) \\ 0 \\ \sin(\alpha) \end{bmatrix}}^{\boldsymbol{v}_{\infty,j}}_{\boldsymbol{v}_a} + \underbrace{\begin{bmatrix} u_{t,j} \\ 0 \\ w_{t,j} \end{bmatrix}}_{\boldsymbol{v}_{t,j}} + \begin{bmatrix} 0 \\ 0 \\ w_{j,c/4} \end{bmatrix} \tag{37}$$





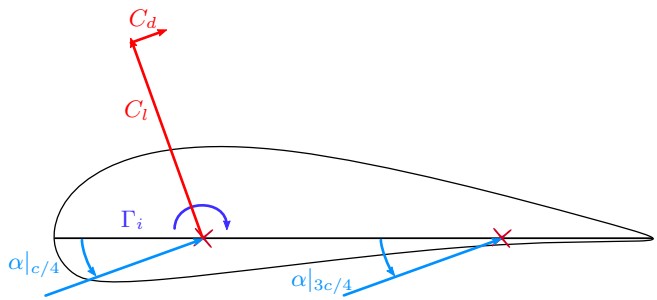

**Figure 4.** Angles of attack evaluation for an airfoil.

The velocity induced by the trailed vorticity is

$$w_{j,c/4} = \sum_{p=1}^{N_p+1} \frac{\gamma_p}{4\pi} \int_{-\infty}^{0} \frac{y_j - y_p}{r_j{}^3} dx = \sum_{p=1}^{N_p+1} \frac{\gamma_p}{4\pi} \frac{1}{y_j - y_p}. \tag{38}$$

The angle of attack used to project the lift and drag is then

$$\alpha_{j,c/4} = \arctan\left( \frac{\boldsymbol{v}_{r,j,c/4}(3)}{\boldsymbol{v}_{r,j,c/4}(1)} \right) + \beta_j. \tag{39}$$

The nonlinear problem formulated to find the induced velocities at each aerodynamic section, and thus the aerodynamic coefficients, is solved iteratively by imposing

$$\Gamma_j \frac{v_{\infty,j}}{v_{r,j}^2} = \frac{c_j C_{l,j}}{2} \tag{40}$$

at each evaluation point.

The lift force at each aerodynamic section is

$$l_j = \frac{1}{2} \rho c_j C_{l,j} v_{r,j}^2, \tag{41}$$

and the wing lift coefficient is defined as

$$C_L = \frac{L}{\frac{1}{2}\rho A v_a^2} = \frac{\sum_{j=1}^{N_p} c_j C_{l,j} v_{r,j}^2 dy_j}{A v_a^2}. \tag{42}$$

The integral induced drag $D_i$ can be computed from the induced drag at each wing section $d_{i,j}$, which correspond to the local lift $l_j$ times the local induced angle of attack $\alpha_{i,j,c/4} = \arctan\left(\frac{w_{j,c/4}}{\boldsymbol{v}_{r,j,c/4}(1)}\right)$. The induced drag coefficient as

$$C_{D,i} = \frac{D_i}{\frac{1}{2}\rho A v_a^2} = \frac{\sum_{j=1}^{N_p} c_j C_{l,j} \alpha_{i,j,c/4} v_{r,j}^2 dy_j}{A v_a^2}. \tag{43}$$

Finally, the three-dimensional profile drag coefficient due to the airfoils $C_{D,a}$ can be computed as

$$C_{D,a} = \frac{\sum_{j=1}^{N_p} c_j C_{d,j} v_{r,j}^2 dy_j}{A v_a^2}, \tag{44}$$

where $C_{d,j}$ is the airfoil drag coefficient.





## 2.4 Windplane wake model

To conclude the modeling of the windplane aerodynamics, we need to find how much the wind is slowed down, which means finding the induction $a_f$ in Fig. 2. Trevisi et al. (2023b) show that for large turning radius $\left(\frac{b/2}{R_0}\right)^2 << 1$ and large wing speed ratios $\frac{1}{\lambda^2} << 1$, the aerodynamic induction $a$ generated by the helical vortex system (Fig. 5(a)) can be modeled as the sum of two terms: one due to the first half rotation of the helical vortex filaments (the near wake induction $a_n$) and one related to the semi-infinite cascade of vortex rings (the far wake induction $a_f$) as in Fig. 5(b). The near wake induction, or the near wake induced drag $D_i$, can be computed by assuming straight semi-infinite trailed vortices, as carried out in Sect. 2.3.

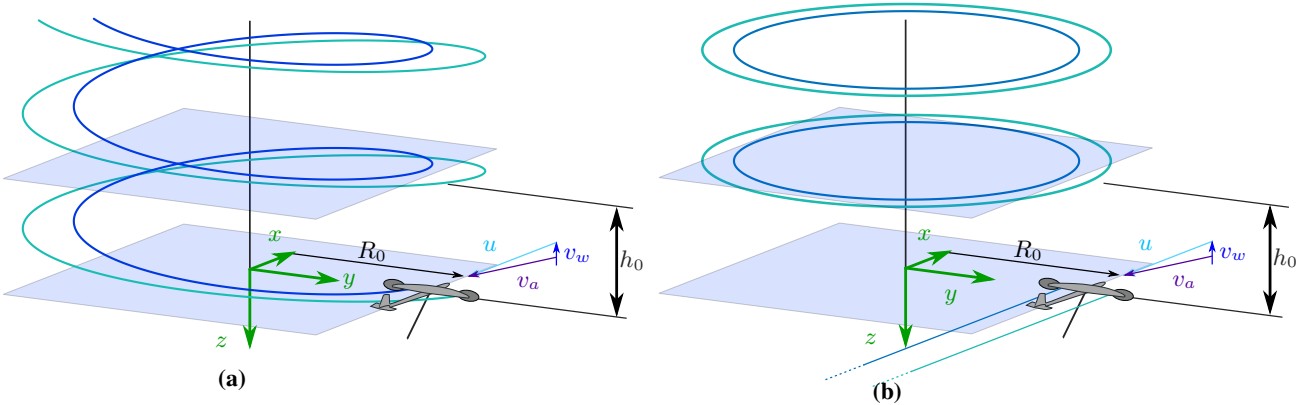

**Figure 5.** Left and right rolled-up trailed helicoidal vortices (a) and relative modeling after assumptions (b) (Trevisi et al. (2023b)).

The far wake induction $a_f$ can be instead computed by summing up the contribution of each vortex ring. Their strength is equal to the tip vortex strength

$$\Gamma_0 = \int_0^{b/2} \gamma\, dy, \tag{45}$$

and their spanwise position is

$$y_v = \frac{1}{\Gamma_0} \int_0^{b/2} \gamma\, y\, dy. \tag{46}$$

The wake pitch is

$$h_0 = v_w \left(1 - a_r\right) \frac{2\pi R_0}{u}, \tag{47}$$

where $v_w \left(1 - a_r\right)$ is the axial velocity of the vortex rings. Gaunaa et al. (2020) evaluate the axial velocity of the vortex rings by modeling them as infinite straight vortices and assuming that the dominant effect in the motion of the vortices is the vorticity





of the nearest vortex. With this model, tested to be valid by Trevisi et al. (2025), the convection velocity of the rings is

$$a_r = \frac{1}{v_w} \frac{\Gamma_0}{2\pi(2y_v)}.$$

(48)

The induction due to the far wake $a_f$ (Fig. 2) can now be evaluated with (Trevisi et al. (2023b))

$$a_f = \frac{1}{v_w} \frac{\Gamma_0}{4\pi y_v} \sum_{k=1}^{\infty} \left( \Upsilon_k\left(\eta = -\eta_v, \lambda_0\right) + \Upsilon_k\left(\eta = \eta_v, \lambda_0\right) \right) = a_r \sum_{k=1}^{\infty} \left( \Upsilon_k\left(\eta = -\eta_v, \lambda_0\right) + \Upsilon_k\left(\eta = \eta_v, \lambda_0\right) \right),$$

(49)

where $\eta_v = \frac{y_v}{R_0}$ is the tip vortex normalized position and $\lambda_0 = \frac{2\pi R_0}{h_0} = \frac{\lambda}{(1-a_r)}$ is the normalized torsional parameter of the helicoidal wake. The shape factor $\Upsilon_k(\eta, \lambda_0)$ models the induction of one vortex ring, and can be computed in closed form as (Trevisi et al. (2023b))

$$\Upsilon_k(\eta, \lambda_0) = \frac{-2\eta}{\left(\eta^2 + \left(\frac{2\pi k}{\lambda_0}\right)^2\right)^{1/2}} \left( F(f) + \frac{\eta(\eta-2) - \left(\frac{2\pi k}{\lambda_0}\right)^2}{(\eta-2)^2 + \left(\frac{2\pi k}{\lambda_0}\right)^2} E(f) \right),$$

(50)

where $F(f)$ and $E(f)$ are the complete elliptic integral of the first and second kind respectively and $f = \frac{4(\eta-1)}{\eta^2 + \left(\frac{2\pi k}{\lambda_0}\right)^2}$.

## 3 Windplane aerodynamic design problem

In this section, we formulate the aerodynamic design problem formulation and we explore the optimal design space. The optimal aerodynamic design problem reads

$$\max_{\boldsymbol{x} = (a_f, \boldsymbol{\Gamma}, \alpha, c_0, tr, \lambda_t, K_t)} C_P(\boldsymbol{x}, \overbrace{\underbrace{b, m, N_{bl}, R_{t0}, R_t}_{\text{turbines}}, \underbrace{C_{d,te}, D_{te}, L_{te}}_{\text{tether}}, \underbrace{C_l(\alpha), C_d(\alpha)}_{\text{airfoil}}}^{\text{parameters: } \Theta_p})$$

$$\text{subject to: } h_{a_f}(\boldsymbol{x}, \Theta_p) = 0 \qquad \text{far wake eq. constraint}$$

$$\boldsymbol{h}_\Gamma(\boldsymbol{x}, \Theta_p) = \boldsymbol{0} \qquad \text{angles of attack eq. constraints}$$

(51)

where the optimization variables $\boldsymbol{x}$ are the induction due to the far wake $a_f$, the bound circulation at each wing aerodynamic section $\boldsymbol{\Gamma}$, the wing angle of attack $\alpha$, the chord at the wing center $c_0$, the taper ratio of the trapezoidal wing $tr$, the onboard turbine tip speed ratio $\lambda_t$ and the parameter $K_t$, determining the radial distribution of $k(r_t)$ (see Sect. 2.2). The induction due to the far wake $a_f$ is settled by the optimizer to satisfy $h_{a_f}(\boldsymbol{x}, \Theta_p) = 0$ (Eq. (49)) and the bound circulation at each aerodynamic section $\Gamma_j$ is settled to satisfy $\boldsymbol{h}_\Gamma(\boldsymbol{x}, \Theta_p) = \boldsymbol{0}$ (Eq. (40)).

The objective function is the power coefficient $C_P$ (Eq. 15). Optimizing for this power coefficient is equivalent to optimizing for the shaft power, while keeping the wing span constant.

A trapezoidal wing with constant twist is here analyzed. Future studies will investigate the influence of different wing shapes and twist distributions on the system performances. The onboard turbines are conceptually designed by modifying their





tip speed ratio $\lambda_t$ and the parameter $K_t$. The radial distribution of the parameter $k(r_t)$ (see Sect. 2.2) is assumed parabolic and function of $K_t$ as

$$k(r_t) = K_t - \frac{4K_t}{(R_t - R_{t0})^2}\left(r_t - \frac{(R_t + R_{t0})}{2}\right)^2,\qquad (52)$$

where $R_t$ the turbine radius and $R_{t0}$ hub radius. Future studies will investigate the effect of different loading shapes (i.e.
different distribution of the parameter $k(r_t)$) on the system performances.

The other fixed parameters are the wing span $b$, the system mass $m$, the number of turbine blades $N_{bl}$, the tether section drag coefficient $C_{d,te}$, the tether diameter $D_{te}$, the tether length $L_{te}$ and the airfoil polars as a function of the angle of attack $(C_l(\alpha), C_d(\alpha))$.

By solving this optimization problem, we can get to a design of the wing and of the turbines. To get to the chord and
twist of the turbine blades, the Glauert tip correction is applied (Branlard (2017)) [4]. The turbine design features a constant lift coefficient along the blade span at the design tip speed ratio. In order to achieve a realistic design, the design lift coefficient at the blade tip is lowered to delay stall at lower tip-speed ratios and the chord at the blade root is increased to allow for the hub connection. For these reasons, the chord at the root and at the tip is slightly widened, while the twist is adjusted to keep the same lift force.

## 3.1 Design space exploration

We first explore the optimal design space to study the influence of the turbine position and of the airfoil characteristics on the power output. We chose the parameters in Table 1, which are slightly adjusted with respect to the analyses by Trevisi (2024). The wing is discretized into 100 elements of constant size. The optimization problem (51) is solved in MATLAB with the sequential quadratic programming algorithm implemented in the function *fmincon*. The solution of the problem takes a few
tens of seconds on a standard laptop. Local minima, which are not interesting for this application, can be found when the solver finds solutions with angle of attack above stall.

**Table 1.** Parameters for the optimal design problem.

| | | | |
|---|---|---|---|
| $b = 10$ m | $N_{bl} = 3$ | $R_{t0} = 0.2$ m | $R_t = 1.0$ m |
| $m = 100$ kg | $L_{te} = 150$ m | $D_{te} = 12.5$ mm | $C_{d,te} = 0.8$ |

For this study, we compared three turbines' positions (Fig. 6):

a) Turbines rotating inboard down located at one rotor hub radius outward the wing tip $(y_t = \pm\left(\frac{b}{2} + R_{t0}\right) = \pm 5.2$ m$)$ (indicated with continuous lines in the following figures −).

---

[4] $F = \frac{2}{\pi}\arccos\left(\exp\left[-\frac{N_{bl}}{2}\left(1 - \frac{\lambda_{r,t}}{\lambda_t}\right)\sqrt{1 + \lambda_t^2\left(\frac{1 + a_t'(r_t)}{1 - a_t(r_t)}\right)^2}\right]\right)$





b) Turbines rotating inboard down located in front of the wing ($y_t = \pm\frac{b}{4} = \pm 2.5$ m) (indicated with dashed lines in the following figures $--$).

   c) Turbines mounted on pylons (indicated with dotted lines in the following figures $\cdots$). To model this case, we switch off the interactional aerodynamics between the turbines and the wing, while neglecting the parasite drag associated to the pylons.

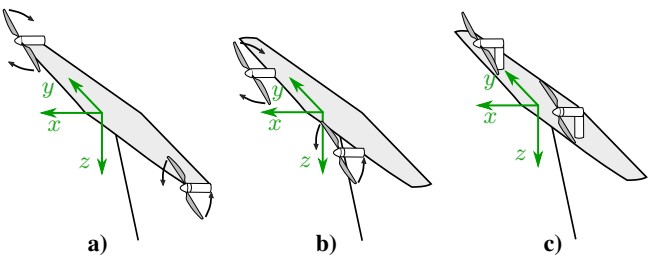

**Figure 6.** Representation of the three windplane layouts considered for the design space exploration study.

Moreover, we consider idealized airfoils on the windplane wing. This is to study the influence of their key metrics on the resulting design. The idealized airfoils have prescribed maximum efficiency $E_{a,max} = \frac{C_l}{C_d}|_{max} = [75, 100, 125]$ at different lift coefficient $C_l|_{E_{a,max}}$. The airfoil lift coefficient is modeled to vary linearly with respect to the angle of attack as $C_l = 2\pi\alpha + 0.3$ and the efficiency $E_a$ is modeled as a parabola $E_a = E_{a,max} - e_{2,a}\left(\left(\alpha - \alpha|_{E_{a,max}}\right)\frac{180}{\pi}\right)^2$, with the peak equal to $E_{a,max}$, achieved at the angle of attack $\alpha|_{E_{a,max}}$ and at the lift coefficient $C_l|_{E_{a,max}}$ [5].

Figure 7 shows the wing lift coefficient $C_L$ as a function of the airfoil lift coefficient of maximum efficiency $C_l|_{E_{a,max}}$. Recall that the data in the figure represent the solutions of the optimization problem (51). The trends show that it is optimal to design the wing such that it operates the airfoils at their maximum efficiency. This is in agreement with the finding by Trevisi (2024).

   Figure 8 shows the optimal aspect ratio $AR$ as a function of the airfoil lift coefficient of maximum efficiency $C_l|_{E_{a,max}}$.

The aspect ratio has a similar physical meaning to the solidity of conventional turbines. For increasing $C_l|_{E_{a,max}}$, increasing aspect ratio are optimal. The optimal aspect ratio is almost insensible to the airfoil maximum efficiency $E_{a,max}$ and to the interactional aerodynamics.

---

[5]The angle of attack $\alpha$ is measured in radians. $e_{2,a} = 1$ for $E_{a,max} = 75$, $e_{2,a} = 1.5$ for $E_{a,max} = 100$, and $e_{2,a} = 2$ for $E_{a,max} = 125$.



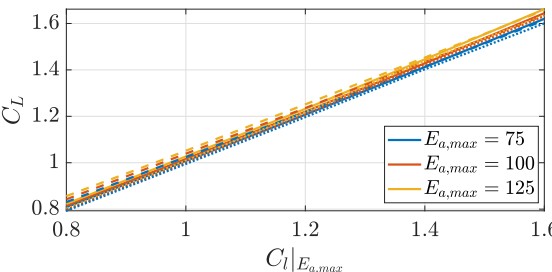

**Figure 7.** Optimal wing lift coefficient as a function of the lift coefficient of maximum airfoil efficiency $C_l|_{E_{a,max}}$. Trends shown for three different values of maximum airfoil efficiency $E_{a,max}$ and three turbines position (on the pylons $(\cdots)$, in front of the wing $(--)$ and on the wingtip $(-)$).

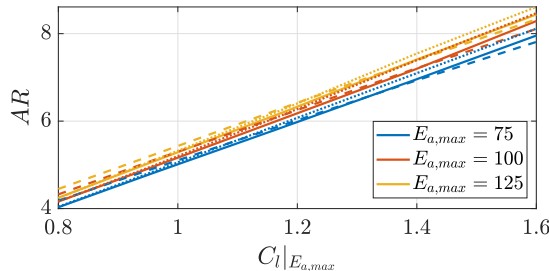

**Figure 8.** Optimal wing aspect ratio as a function of the lift coefficient of maximum airfoil efficiency $C_l|_{E_{a,max}}$. Trends shown for three different values of maximum airfoil efficiency $E_{a,max}$ and three turbines position (on pylons $(\cdots)$, in front of the wing $(--)$ and on the wingtip $(-)$).

Figure 9 shows the optimal taper ratio as a function of the airfoil lift coefficient of maximum efficiency $C_l|_{E_{a,max}}$. Higher taper ratios are preferable if the turbines are placed at the wing tips. This is because higher taper ratio wings have more lifting 330 area behind the turbines, enhancing their beneficial effect on the power production.

Figures 10, 11 and 12 shows the onboard turbine tip speed ratio $\lambda_t$, the turbines power coefficient $C_{P,t}$ and ratio between power and thrust coefficient $C_{P,t}/C_{T,t}$. Recall that the onboard turbines are designed to maximize the windplane power coefficient $C_P$ (Eq. 15) and not their own power coefficients $C_{P,t}$. Considering the interactional aerodynamics, the optimal tip speed ratio decreases to increase the wake swirl and thus to increase the change in inflow angle, reducing the induced drag in 335 the wing aerodynamic sections behind the rotors. With the turbines placed at the wing tips, the tip speed ratio is the lowest, leading to lower $C_{P,t}/C_{T,t}$. Designs with lower tip speed ratios are typically preferable to reduce the noise emission and the erosion.

Figure 13 shows the wing speed ratios for the analyzed cases. The windplane flies slightly slower when the turbines are placed in front of the wing.

WIND
ENERGY
SCIENCE
DISCUSSIONS

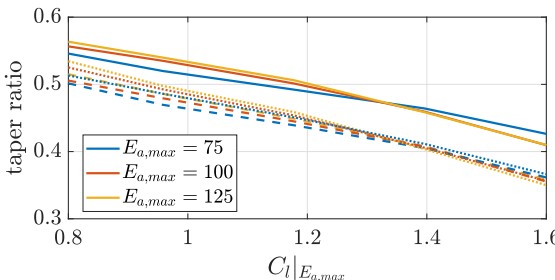

**Figure 9.** Optimal taper ratio as a function of the lift coefficient of maximum airfoil efficiency $C_l|_{E_{a,max}}$. Trends shown for three different values of maximum airfoil efficiency $E_{a,max}$ and three turbines position (on pylons ($\cdots$), in front of the wing ($--$) and on the wingtip ($-$)).

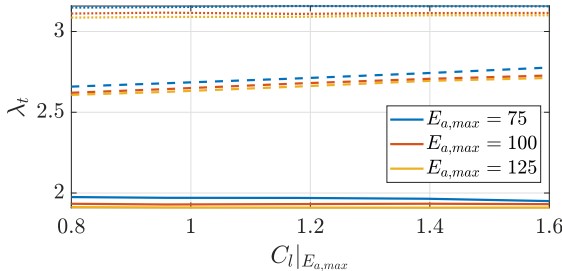

**Figure 10.** Optimal onboard turbines tip speed ratio as a function of the lift coefficient of maximum airfoil efficiency $C_l|_{E_{a,max}}$. Trends shown for three different values of maximum airfoil efficiency $E_{a,max}$ and three turbines position (on pylons ($\cdots$), in front of the wing ($--$) and on the wingtip ($-$)).

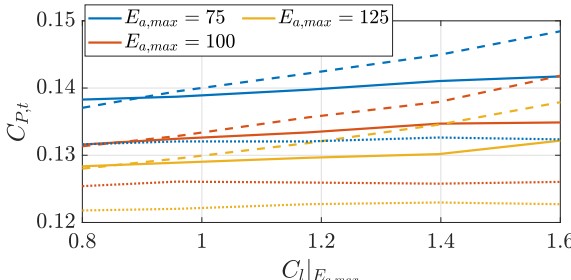

**Figure 11.** Optimal onboard turbine power coefficient as a function of the lift coefficient of maximum airfoil efficiency $C_l|_{E_{a,max}}$. Trends shown for three different values of maximum airfoil efficiency $E_{a,max}$ and three turbines position (on pylons ($\cdots$), in front of the wing ($--$) and on the wingtip ($-$)).

Figure 14 shows the optimal power coefficient $C_P$ as a function of the airfoil lift coefficient of maximum efficiency $C_l|_{E_{a,max}}$. The power coefficient is sensitive to the maximum airfoil efficiency $E_{a,max}$ and slightly sensitive to $C_l|_{E_{a,max}}$.



WIND
ENERGY
SCIENCE
DISCUSSIONS

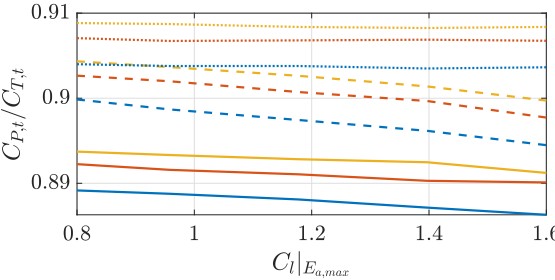

**Figure 12.** Optimal onboard turbine power coefficient to thrust coefficient $C_{P,t}/C_{T,t}$ as a function of the lift coefficient of maximum airfoil efficiency $C_l|_{E_{a,max}}$. Trends shown for three different values of maximum airfoil efficiency $E_{a,max}$ and three turbines position (on pylons $(\cdots)$, in front of the wing $(--)$ and on the wingtip $(-)$).

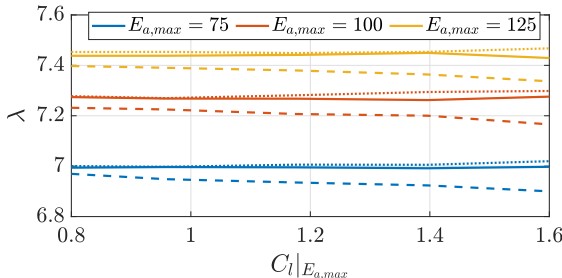

**Figure 13.** Optimal windplane wing speed ratio as a function of the lift coefficient of maximum airfoil efficiency $C_l|_{E_{a,max}}$. Trends shown for three different values of maximum airfoil efficiency $E_{a,max}$ and three turbines position (on pylons $(\cdots)$, in front of the wing $(--)$ and on the wingtip $(-)$).

Indeed, larger power coefficients can be achieved with more efficient airfoils. Moreover, the position of the turbines influence the power production. If they are placed on pylons, the power is the lowest even if the parasite drag associated to the pylons is neglected. Placing the turbines at the wing tips improves the power production compared to placing them in front of the wing.

Note that the turbines rotate inboard down. This can also be seen in Fig. 15, where the position of the turbines $y_t$ is moved along the wing for three different values of maximum airfoil efficiency $E_{a,max}$ with $C_l|_{E_{a,max}} = 1$. If the turbines are placed in front of the wing, the wing sections placed inward the turbine position experience a beneficial change in inflow angle, while the wing sections placed outward experience a disadvantageous change in inflow angle. If the turbines are placed at the wing tips, there are no wing sections outward the turbine position, so that only the beneficial effects in the inward wing sections are

experienced. Since the nacelle is not modeled in this study, the portion of wing behind the rotor hub position sees an slight increase in inflow speed. For this reason, the peak in $C_P$ corresponds to one rotor hub diameter inward. The peak in real systems will be influenced by the presence of the nacelle and the associated parasite drag, so the exact position and amplitude of this peak will change consequently. With the turbines getting closer to the wing tip, the interactional aerodynamics become more beneficial for the power production, and thus it is optimal for the onboard turbines to increase their wake swirl. For this

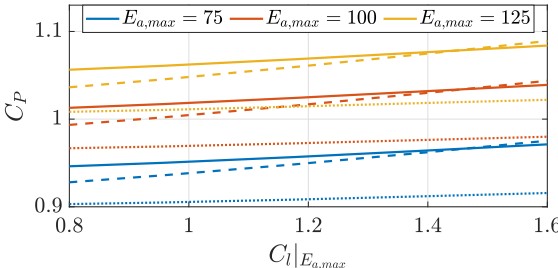

**Figure 14.** Optimal power coefficient as a function of the lift coefficient of maximum airfoil efficiency $C_l|_{E_{a,max}}$. Trends shown for three different values of maximum airfoil efficiency $E_{a,max}$ and three turbines position (on pylons ($\cdots$), in front of the wing ($--$) and on the wingtip ($-$)).

reason, the optimal onboard turbines tip speed ratio decreases for the turbines getting closer to the tip, as shown in Fig. 16. To shed light on how big the beneficial interaction is, the figures show also the effect of rotating the turbines the non-beneficial outboard down direction, where a significant drop in power coefficient can be observed.

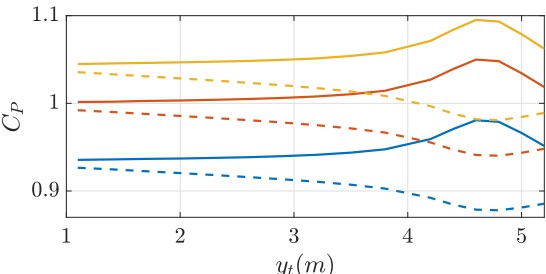

**Figure 15.** Optimal power coefficient as a function of the spanwise position of the turbines for rotation direction inboard down ($-$) and outboard down ($--$). Trends shown for three different values of maximum airfoil efficiency $E_{a,max}$ with $C_l|_{E_{a,max}} = 1$.

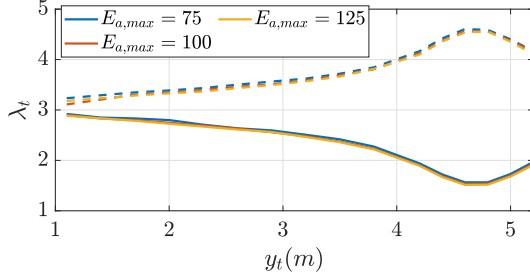

**Figure 16.** Optimal onboard turbines tip speed ratio as a function of the spanwise position of the turbines for rotation direction inboard down ($-$) and outboard down ($--$). Trends shown for three different values of maximum airfoil efficiency $E_{a,max}$ with $C_l|_{E_{a,max}} = 1$.




As a final study, the optimal designs are studied as a function of the onboard turbines' radius. Figure 17 shows the optimal power coefficient as a function of the turbines' radius. The power increases for larger radius, such that the turbines can operate at higher $C_{P,t}/C_{T,t}$ (Fig. 18). Higher $C_{P,t}/C_{T,t}$ indicate that less power is lost in the conversion from thrust power (i.e. the power experienced by the windplane dynamics) to generated power (i.e. the power experienced by the turbines' shaft). When the turbines are placed at the wing tips, a lower $C_{P,t}/C_{T,t}$ is optimal because of the beneficial effects of the interactional aerodynamics. From an aerodynamic point of view, the turbines' radius should be as large as possible. However, the final turbine size will be determined by including structural, electrical and control considerations. In this study, we keep the turbine radius fixed to $R_t = 1$ m, which seems to be reasonable.

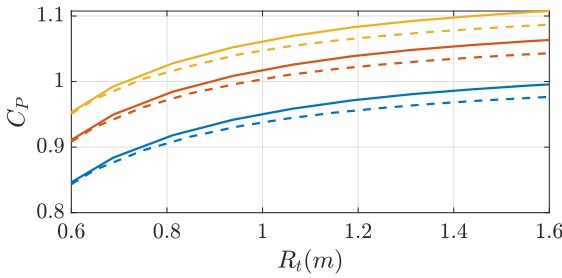

**Figure 17.** Optimal power coefficient as a function of the onboard turbines' radius $R_t$ for two turbines position ( in front of the wing $(--)$ and on the wingtip $(-)$). Trends shown for three different values of maximum airfoil efficiency $E_{a,max}$ with $C_l|_{E_{a,max}} = 1$.

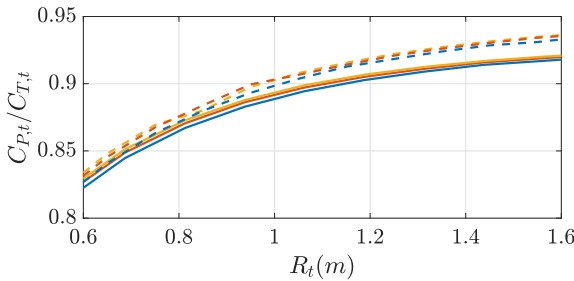

**Figure 18.** Onboard turbines $C_{P,t}/C_{T,t}$ as a function of the onboard turbines' radius $R_t$ for two turbines position (in front of the wing $(--)$ and on the wingtip $(-)$). Trends shown for three different values of maximum airfoil efficiency $E_{a,max}$ with $C_l|_{E_{a,max}} = 1$.

To conclude, this study confirms the thesis (Trevisi (2024)) that conventional efficient airfoils should be used in the design of windplanes and that the optimal aspect ratio is finite. The windplane should then be operated at the lift coefficient corresponding to the airfoil maximum efficiency. The optimal wing taper ratio and the optimal onboard turbines tip speed ratio are influenced by the interactional aerodynamics. If the turbines are placed at the wing tips and the windplane designed accordingly, the power can increase considerably.





# 4 Vortex particle method: DUST

To validate our analytical results, we use DUST[6], an open-source aerodynamic simulation software that employs the vortex particle method (Cottet and Koumoutsakos (2000)). This grid-free Lagrangian approach effectively models free wake vorticity evolution and has been developed in accordance with modern FORTRAN standards. It incorporates classical potential-based aerodynamic models, including the lifting line method (Gallay and Laurendeau (2015); Piszkin and Levinsky (1976)), surface panels (Piszkin and Levinsky (1976)), and vortex lattice elements (Katz and Plotkin (2001)). While the software assumes incompressible potential flow, compressibility effects are accounted for using the Prandtl-Glauert correction in steady aerodynamic load computations. Additionally, lifting line and vortex lattice elements incorporate both compressibility and viscous effects through Mach-dependent aerodynamic coefficient tables.

We select the lifting line element to model the turbines blades because it eliminates the need for explicitly modeling the surface mesh, as the flow particles do not interact directly with the blade surface. For the wing, using a lifting line element can pose challenges in detecting the penetration condition (Niro et al. (2024)). Therefore we opt for the vortex lattice element as it provides accurate results without the higher computational cost associated with surface panel elements.

# 5 Optimal aerodynamic design

In this section, we present the optimal design and analyze it with DUST. The main inputs for this case study are given in Table 1. The 10 m wingspan windplane was initially designed by Trevisi (2024), and its design is here refined. The NACA4421 airfoils are used for both the wing and the onboard turbines. The chord-based Reynolds number for the wing is above 1 million, while the turbine blades have a chord-based Reynolds number between 200'000 and 500'000. The dependence of the airfoils polars on the Reynolds number can be taken into account in DUST by providing the polars for different Reynolds numbers. This airfoil is chosen because it has a high maximum efficiency $E_{a,max} = 103$ ($Re = 10^6$) at lift coefficient of $C_l|_{E_{a,max}} = 1.01$, with a relative large thickness (21%), which is good for the structural design.

The main optimization results are reported in Table 2. The wing is discretized into 60 elements with a cosine distribution to refine the discretizations after the turbines. The windplane achieves a power coefficient of $C_P = 0.99$. The turbines operate with a low tip speed ratio of $\lambda_t = 1.91$. Lowering the tip speed ratio increase the wake swirl and thus the inflow angle at the wing sections after the turbines. This low value of $\lambda_t$ is also good to limit the blade erosion and the acoustic emission. The wing has a low aspect ratio of $AR = 5.1$, which is good for structural design and maneuverability, has a taper ratio of $tr = 0.60$ and operates at the maximum efficiency lift coefficient $C_L = 1.01$, leading to a wing speed ratio of $\lambda = 7.12$. The spanwise efficiency $e_b$ indicates how efficiently the wing generate lift with respect to the induced drag (Anderson (2017)). It can be estimated as

$$e_b = \frac{C_L^2}{\pi AR C_{Di}}.$$ (53)

---

[6]https://www.dust.polimi.it/




Modeling the wing as a lifting line, which is accurate for high aspect ratio, the spanwise efficiency for an elliptic isolated wing is $e_b = 1$. Modeling the wing with the present formulation, which is accurate even for low aspect ratio, the spanwise efficiency of the wing, accounting for the onboard turbine's influence, can take values above one. This value is then representative of the effects of the onboard wind turbines on the wing induced drag. Note that the spanwise efficiency $e_b$ should not be confused

with the Oswald efficiency, which accounts also for the parasite drag components of the windplane (Trevisi (2024); Anderson (2017)) and the induced drag produced by other lifting surfaces on the windplane (e.g. stabilizers).

**Table 2.** Main aerodynamic optimization results.

| | | | | | |
|---|---|---|---|---|---|
| $C_P = 0.99$ | $C_T = 3.19$ | $C_{P,t} = 0.137$ | $C_{T,t} = 0.155$ | $\lambda_t = 1.91$ | $K_t = 0.23$ |
| $c_{root} = 2.43$ m | $tr = 0.60$ | $c_{tip} = 1.47$ m | AR $= 5.12$ | $\alpha = 10.9\,°$ | $a_f = 0.02$ |
| $C_L = 1.01$ | $C_{Di} = 0.059$ | $C_d = 0.011$ | $C_{D,te} = 0.019$ | $\lambda = 7.12$ | $e_b = 1.08$ |

The onboard wind turbines chord and twist are shown in Figure 19. The values are given in Table B1. The twist changes drastically from the inner to the outer sections because the operational tip speed ratio $\lambda_t$ is very low, compared to conventional turbines.

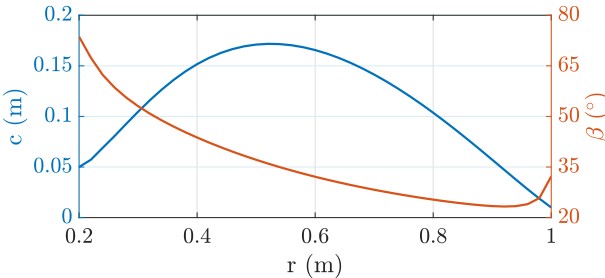

**Figure 19.** Onboard turbine chord $c$ and twist $\beta$ as function of the radius.

## 5.1   Isolated turbine

The onboard turbines are now studied with DUST, and the results compared with the vortex cylinder model. The turbines blades are modeled as lifting lines with 40 elements, according to the convergence study performed in Appendix A and the simulation is run for 8 revolutions with a time step corresponding to $5°$, according with the DUST simulations by Niro et al. (2024). The turbines are simulated at a tip speed ratio of $\lambda_t = 1.91$, finding a thrust coefficient of $C_{T,t} = 0.144$, a power

coefficient of $C_{P,t} = 0.125$ and an efficiency of $C_{P,t}/C_{T,t} = 0.86$. These values are really close to the values found with the vortex cylinder model (Table 2, $C_{T,t} = 0.155$, $C_{P,t} = 0.137$, $C_{P,t}/C_{T,t} = 0.88$). The angle of attack and the corresponding sectional lift coefficients along the blade are shown in Fig. 20.

Figure 21 shows the induction of the turbines predicted by the vortex cylinder model and by DUST at a few distances from the rotor plane for $\lambda_t = 1.91$. The trends show a good agreement at the rotor plane ($0R_t$), with DUST predicting slightly lower

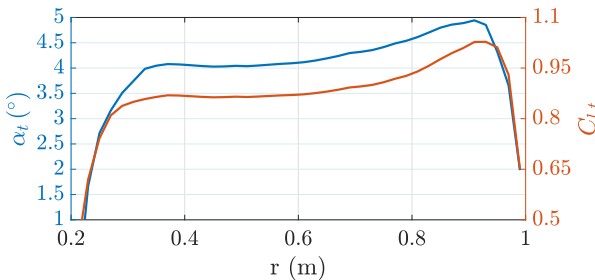

**Figure 20.** Onboard turbines angle of attack $\alpha_t$ and lift coefficient $C_{l,t}$ as a function of the radial coordinate ($\lambda_t = 1.91$).

induced velocities. This could be due to the lower thrust coefficient. The trends after the rotor shows the reduction in wind speed due to the wake expansion.

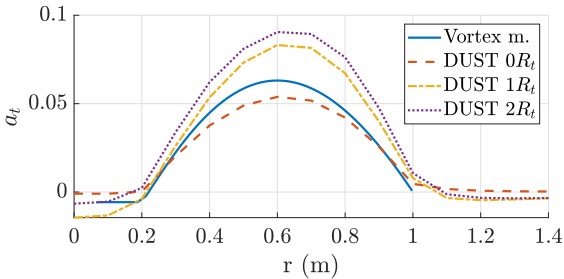

**Figure 21.** Onboard turbines' induction $a_t$ as a function of the radial coordinate evaluated with the vortex cylinder model and with DUST at a few downstream distances ($\lambda_t = 1.91$).

Figure 22 shows the change in angle of attack at different downstream positions. The vortex cylinder model and DUST are in good agreement at the rotor plane, with the vortex cylinder model predicting slightly higher change in angle of attack $\alpha_{i,t}$.

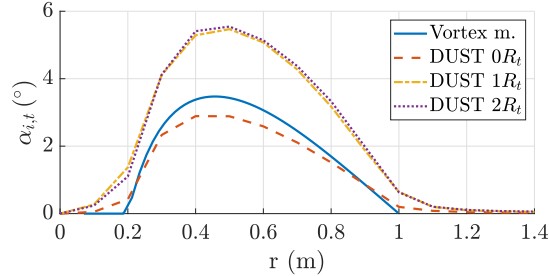

**Figure 22.** Onboard turbines' induced change in angle of attack $\alpha_{i,t}$ as a function of the radial coordinate evaluated with the vortex cylinder model and with DUST at a few downstream distances ($\lambda_t = 1.91$).




To conclude the analysis of the turbine, the performance is studied for different tip speed ratios $\lambda_t$. Figure 23 shows the power and thrust coefficients of the turbines as a function of their tip speed ratio computed with DUST. When the thrust coefficient

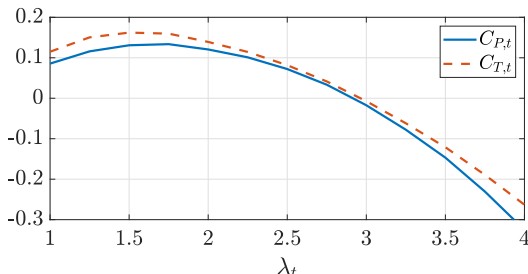

**Figure 23.** Onboard turbines' power coefficient $C_{P,t}$ (-) and thrust coefficient $C_{T,t}$ (- -) as a function of the tip speed ratio $\lambda_t$ computed with DUST.

reaches negative numbers, the turbine is operating as a propeller. This turbine is designed to be fundamentally different from a conventional one for wind energy conversion. Indeed, its maximum power coefficient $C_{P,t} \approx 0.13$ is far from the Betz limit, as it is designed to maximize the windplane power coefficient $C_P$. The design tip speed ratio of $\lambda_t = 1.91$ approximately corresponds to the maximum thrust and power coefficient. This means that these turbines cannot produce higher power, and

thus higher braking force, than their design value. Alborghetti et al. (2025) show how to control the windplane turbines in order to smooth the power output, highlighting the need to almost double the power coefficient with respect to the average value in some parts of the loop. Future works should then modify this design to cope with control requirements.

Figure 24 and 25 show the angle of attack and the lift coefficient along the blades for a tip speed ratio of $\lambda_t = 1.5$ and $\lambda_t = 4$ respectively. For low $\lambda_t$, the outer aerodynamic sections start experiencing positive stall, with the angle of attack exceeding

$10°$. For higher $\lambda_t$, the lift coefficients gets to negative values, meaning that the thrust is generated in the opposite direction. The aerodynamic sections along the blade gets closer to negative stall, which appears at $\alpha < -14°$ (Re = 200'000) or $\alpha < -16°$ (Re = 1'000'000) for the considered airfoil (NACA 4421).

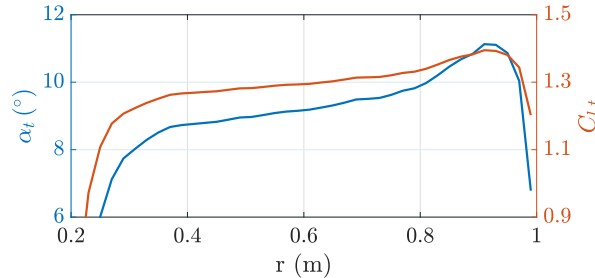

**Figure 24.** Onboard turbines angle of attack $\alpha_t$ and lift coefficient $C_{l,t}$ as a function of the radial coordinate ($\lambda_t = 1.5$).





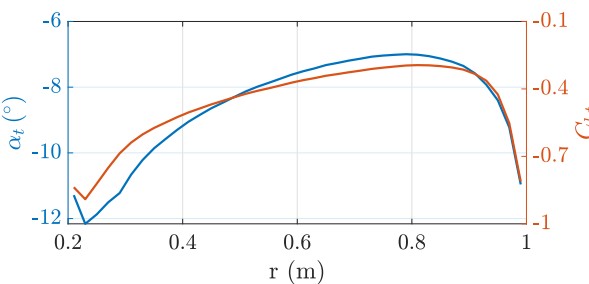

**Figure 25.** Onboard turbines angle of attack $\alpha_t$ and lift coefficient $C_{l,t}$ as a function of the radial coordinate ($\lambda_t = 4$).

## 5.2 Isolated wing

The isolated wing is here studied with DUST and with the Weissinger lifting line model developed for the design. We model

the wing in DUST with with vortex lattice elements. This modelling approach does not allow for an explicit evaluation of the wing sections angle of attack and lift and drag coefficients, which can be recovered from the sectional loads. Moreover, this method only captures the induced drag and not the profile drag. The wing is discretized in 40 elements chord-wise and 60 elements span-wise in DUST, with a cosine distribution to better resolve the aerodynamics close to the wing tips. The lifting line has the same span-wise discretization.

The lift and induced drag coefficients are presented in Fig. 26 and 27, showing a very good matching between the two approaches. The DUST simulation is run with a wing angle of attack of $11°$ and the lifting line is run with a wing angle of attack of $12.2°$ to match the same wing lift coefficient of $C_L = 1.11$. The wing induced drag coefficient estimated with the two methods is $C_{Di} = 0.074$ (DUST) and $C_{Di} = 0.072$ (LL).

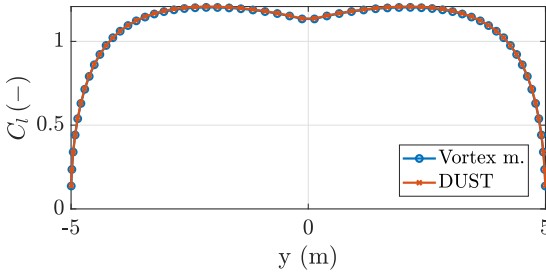

**Figure 26.** Distribution of lift coefficients along the wing span computed with the vortex model (lifting line) and with DUST.





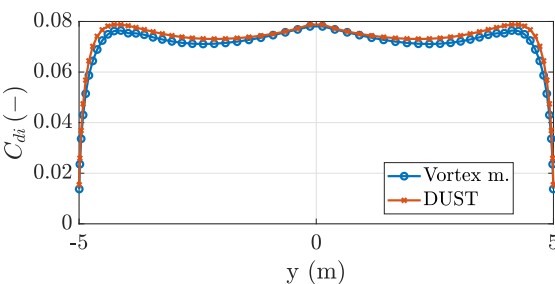

**Figure 27.** Distribution of induced drag coefficients along the wing span computed with the vortex model (lifting line) and with DUST.

### 5.3 Interactional aerodynamics

In this section, the interactional aerodynamics between the wind turbines, rotating inboard down, and the wing is studied. The complete problem is analyzed with DUST and with the interactional vortex model proposed in this paper. Figure 28 and 29 show the loads perpendicular to the inflow (lifting forces) and parallel to the inflow (induced drag forces) estimated with the two methods. The trends of DUST are averaged over the last rotor revolution.

The DUST simulation is run with a wing angle of attack of $11°$ at $v_a = 50$ m/s, and the tip turbines are mounted with a tilt
angle of $-11°$, such that the inflow is perpendicular to the rotor disk. The simulation is run for 8 rotor revolutions with a time step equivalent to a rotor angle of $5°$.

The vortex model is run with a wing angle of attack of $12.5°$ at $v_a = 50$ m/s to match the same integral lifting force computed with DUST of $L = 32.86$ kN. The wing induced drag computed with DUST is $D_i = 2121$ N, while with the vortex model $D_i = 2072$ N. The wing spanwise efficiency computed with DUST (Eq. 53) is $e_b = 1.058$, similar to the value computed
with the vortex model $e_b = 1.080$. The wind turbines in DUST have a thrust coefficient of $C_{T,t} = 0.148$ and a power coefficient of $C_{P,t} = 0.129$ , while in the vortex model they have the values shown in Table 2 ($C_{T,t} = 0.155, C_{P,t} = 0.137$).

The very good match between the vortex model and the DUST simulations proves that the vortex model developed in Sect. 2 captures the main physics of the interactional aerodynamics and then is suitable for design and analysis.

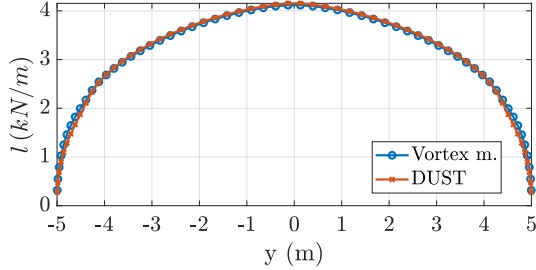

**Figure 28.** Aerodynamic loads acting on the wing perpendicular to the inflow -lifting forces- estimated with the vortex model and with DUST.



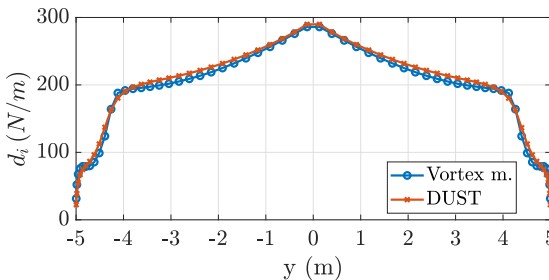

**Figure 29.** Aerodynamic loads acting on the wing parallel to the inflow - induced drag forces - estimated with the vortex model and with DUST.

Figure 30 shows the DUST setup for this simulation. The red dots represent the vortex particles trailed by the lifting surfaces.

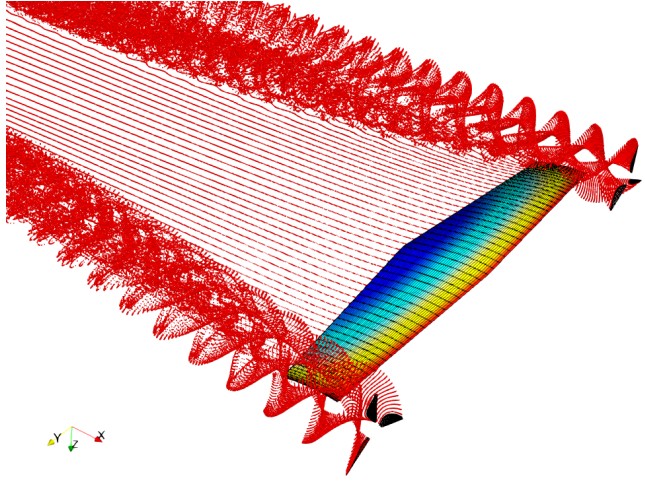

**Figure 30.** Visualization of the DUST simulations. The red dots are the vortex particles.

### 5.4 Analysis out of the design point

To conclude, we analyze the windplane out of the design point. In this study, the wing aerodynamics is exclusively influenced by the wing angle of attack $\alpha$, while the turbines aerodynamics is exclusively influenced by the onboard turbines tip speed ratio $\lambda_t$. Fig. 31 shows the windplane power coefficient $C_P$ (Eq. 15) as a function of $\alpha$ and $\lambda_t$ and is produced with the isolated wing polars (computed with DUST) and the isolated turbines $C_{P,t}, C_{T,t}$ curves (computed with DUST), thus neglecting the aerodynamic interaction between turbines and wing. The far wake induction is computed by estimating the tip vortices strength $\Gamma_0$ (Eq. 45) and their position $y_v$ (Eq. 46) from the local lift distribution in DUST. To include the interactional aerodynamics, we run DUST simulations for different combinations of $\alpha$ and $\lambda_t$ and we evaluate the windplane performances. Figure 32 shows the windplane power coefficient $C_P$ (Eq. 15). The interactional aerodynamics is responsible for an increase in power production

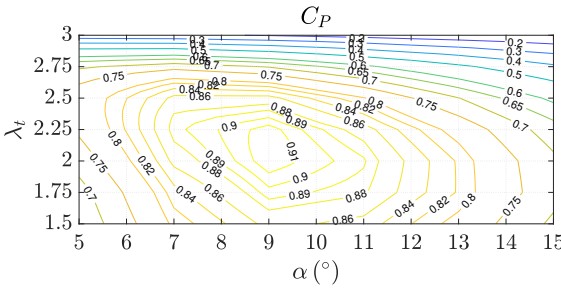

**Figure 31.** Power coefficient $C_P$ as a function of the wing angle of attack $\alpha$ and the turbines tip speed ratio $\lambda_t$ neglecting the aerodynamic interaction between turbines and wing.

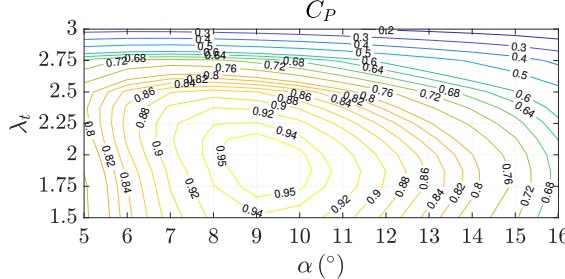

**Figure 32.** Power coefficient $C_P$ as a function of the wing angle of attack $\alpha$ and the turbine tip speed ratio $\lambda_t$ evaluated with DUST.

from $C_P = 0.91$ to $C_P = 0.95$. While the power coefficient $C_P$ informs about the power production of the windplane, the
thrust coefficient $C_T$, in Fig. 33 (Eq. 16), informs about the aerodynamic force generated by the windplane. These two figures can be used to plan the high-level control of the windplane. Indeed, the windplane should be operated at the maximum power coefficient to maximize power production. To reduce power production, the control system might prefer to reduce the angle of attack $\alpha$ instead of increasing the tip speed ratio $\lambda_t$, as this direction decreases the $C_T$ values, then reducing loads.

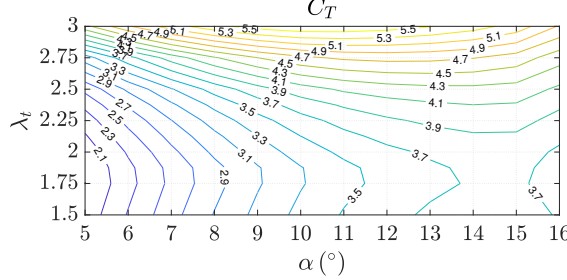

**Figure 33.** Thrust coefficient $C_T$ as a function of the wing angle of attack $\alpha$ and the turbine tip speed ratio $\lambda_t$ evaluated with DUST.



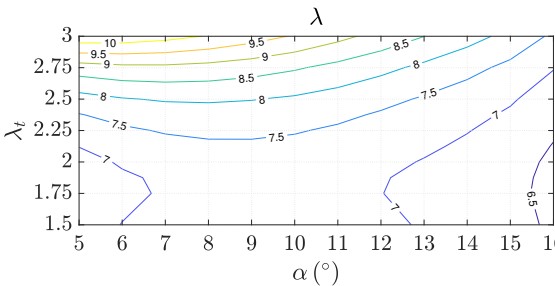

**Figure 34.** Wing speed ratio $\lambda$ as a function of the wing angle of attack $\alpha$ and the turbines tip speed ratio $\lambda_t$ evaluated with DUST.

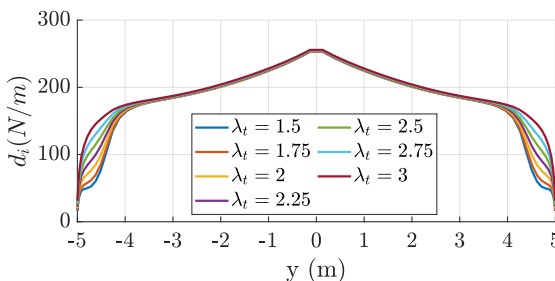

**Figure 35.** Spanwise distribution of induced drag $d_i$ for different onboard turbine tip speed ratios $\lambda_t$.

To conclude the analyses of the design, Figure 34 shows the wing speed ratio, informing about the windplane speed $u$, as a function of $\alpha$ and $\lambda_t$. Clearly, the turbines have the strongest influence on the wing speed ratio, as they act as a brake. It can be also noted that, for a given $\lambda_t$, $\lambda$ has a maximum. This plot can be then used to plan and control the windplane speed. Finally, Fig. 35 shows the induced drag as a function of the onboard turbines tip speed ratio for $\alpha = 10°$. For low $\lambda_t$, the turbines swirl contributes largely to the reduction of the induced drag.

## 6 Conclusions

In this paper, we develop a methodology for the aerodynamic design and analysis of windplanes (the aircraft of fly-gen Airborne Wind Energy Systems), we verify the methodology by comparison with higher fidelity models and characterize the windplane design further with the vortex particle method.

In the method, a novel engineering model for the onboard turbines aerodynamics, the wing aerodynamics and their interactional aerodynamics is employed and coupled to a windplane steady-state model and a windplane far wake model. The turbines are modeled as vortex cylinders, while the wing is modeled with a lifting line method, accounting for the turbines induced velocities. An optimization problem is formulated to concurrently design the turbines and the wing, with a constant twist and a trapezoidal planform. The optimization objective is the windplane power coefficient, which uses as reference area a disk with radius the wingspan. This is equivalent to take as objective the power while keeping the wingspan fixed.





Using the proposed approach, a design space exploration study is carried out to study the dependence of the optimal design with respect to the turbines position and to the airfoil characteristics. We find that placing the turbines at the wing tips and rotating them inboard down lead to higher power production compared to mounting them on pylons (i.e. no interactional aerodynamics) or placing them in front of the wing. The interactional aerodynamics between the onboard turbines and the wing mainly arises because the turbines wake swirl modifies the inflow angle in the following wing aerodynamic sections. By choosing the correct rotor direction, inboard down, and placing the turbines at the wing tips, the outer part of the wing experiences an increase in inflow angles and thus an increase in performances. To enhance this beneficial effect, the optimal wing taper ratio is larger and the optimal onboard turbines tip speed ratio smaller compared to the cases with the turbines placed elsewhere. The design space exploration study also reveals that conventional efficient (high $C_l/C_d$) airfoils should be used in the design of windplanes, confirming the results of Trevisi (2024). The wing aspect ratio is then designed such that the power is maximized when the wing, with constant twist, operates at the lift coefficient correspondent to the airfoil maximum efficiency.

Later, NACA4421 airfoils are considered for the aerodynamic optimization of the wing and of the turbines, placed at the wing tips. The turbines are characterized by an optimal tip speed ratio of $\lambda_t = 1.91$, which dictates their twist and chord distribution. The wing is designed with low aspect ratio ($AR = 5.1$) and to operate at the lift coefficient correspondent to the airfoil maximum efficiency. The vortex models of the isolated turbines, isolated wing and their aerodynamic interaction are compared with the solution of the lifting line, the vortex lattice method and the vortex particle method implemented in open-source code DUST, finding very good agreement. The behavior of the turbines is studied with DUST as a function of the tip speed ratio. Finally, the windplane is studied with DUST at different angles of attack $\alpha$ and at different turbines tip speed ratios $\lambda_t$, finding the maximum power coefficient $C_P = 0.95$ at $\alpha = 9°$ and $\lambda_t = 2$.

This paper describes a comprehensive theory of the windplane aerodynamics, allowing for an in-depth understanding of the main aerodynamic phenomena. This work will be used as a baseline for more detailed aerodynamic studies on the turbines, wing and the tail. It could be also used for computing the aerodynamic derivatives to be used in flight stability analyses, for the closed-loop control development, for the structural design and for airborne wind farm studies.



## Nomenclature

**Latin Symbols**

$A$      Wing area

$a_f$      Induction due the far wake

$a_r$      Normalized axial velocity of the far wake vortex rings

$AR$      Wing aspect ratio

$a_t$      Induction of the onboard turbine

$b$      Wing span

$C_D$      System drag coefficient

$C_{D,i}$      Induced wing drag coefficient

$C_{d,i}$      Sectional induced drag coefficient

$C_{D,a}$      Wing drag coefficient due to the airfoils

$C_{D,te}$      Equivalent tether drag coefficient

$C_{d,te}$      Drag coefficient of the tether section

$C_L$      Wing lift coefficient

$C_P$      Windplane power coefficient

$C_{P,t}$      On-board wind turbine power coefficient

$C_T$      Windplane thrust coefficient

$C_{T,t}$      On-board wind turbine thrust coefficient

$D_{te}$      Tether diameter

$E$      Windplane aerodynamic efficiency, including the tether and the turbines thrust

$e_b$      Spanwise efficiency

$L_{te}$      Tether length

$m$      Windplane mass plus one third of the tether mass

$R_0$      Turning radius

$R_t$      Onboard turbines radius

$R_{t0}$      Onboard turbines hub radius

$tr$      Taper ratio

$u$      Windplane tangential velocity

$v_a$      Apparent wind speed in the near wake

$v_w$      Wind speed

**Greek Symbols**

$\alpha$      Wing angle of attack, defined as the angle between the x direction and the apparent wind speed $v_a$

$\alpha_{i,t}$      Induced change of angle of attack in the wing due to the turbine swirl

$\gamma_n$      Inflow angle in the near wake

$\lambda$      $u/v_w$: wing speed ratio

$\lambda_t$      Onboard turbines tip speed ratio

$\Phi$      Opening angle of the cone swept by the windplane during one loop

$\rho$      Air density

$\xi_t$      $R_t/(b/2)$ Normalized onboard turbines radius





## Appendix A: Turbine convergence study

To select the number of lifting line elements for the turbines for the simulations in DUST, we conduct a convergence study on
an earlier turbine design, similar to the final design. For the turbines, we set a number of elements $N_{el}$ and gradually increased
them until the results converged, as shown in Fig. A1 and Fig. A2, leading to the use of 40 elements for the blade lifting line.
For a good convergence, the simulation is run for at least 6 revolutions (Niro et al. (2024)).

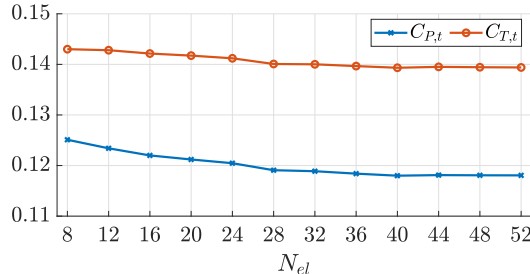

**Figure A1.** Power coefficient $C_{P,t}$ and thrust coefficient $C_{T,t}$ as a function of the lifting line number of elements.

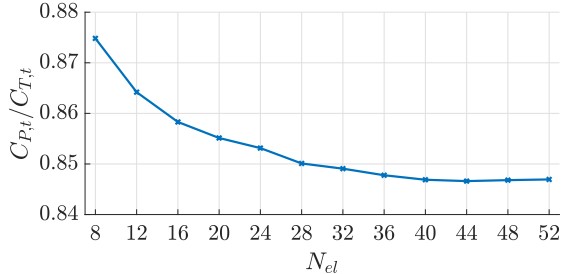

**Figure A2.** Turbine $C_{P,t}/C_{T,t}$ as a function of the lifting line number of elements.

## Appendix B: Turbine data





**Table B1.** Onboard turbine chord and twist distribution for $R_t = 1$ m.

| $r$ (m) | 0.20 | 0.22 | 0.24 | 0.26 | 0.28 | 0.30 | 0.32 | 0.34 | 0.36 | 0.38 | 0.40 | 0.42 | 0.44 | 0.46 |
|---------|------|------|------|------|------|------|------|------|------|------|------|------|------|------|
| $c$ (m) | 0.050 | 0.057 | 0.069 | 0.081 | 0.093 | 0.105 | 0.116 | 0.127 | 0.136 | 0.144 | 0.152 | 0.158 | 0.163 | 0.167 |
| $\beta°$ | 73.65 | 67.41 | 62.33 | 58.59 | 55.58 | 53.04 | 50.82 | 48.83 | 47.01 | 45.32 | 43.75 | 42.27 | 40.87 | 39.56 |
| $r$ (m) | 0.48 | 0.50 | 0.52 | 0.54 | 0.56 | 0.58 | 0.60 | 0.62 | 0.64 | 0.66 | 0.68 | 0.70 | 0.72 | 0.74 |
| $c$ (m) | 0.169 | 0.171 | 0.172 | 0.171 | 0.170 | 0.168 | 0.166 | 0.162 | 0.158 | 0.153 | 0.147 | 0.141 | 0.135 | 0.128 |
| $\beta°$ | 38.32 | 37.14 | 36.03 | 34.97 | 33.97 | 33.02 | 32.11 | 31.26 | 30.45 | 29.68 | 28.95 | 28.26 | 27.60 | 26.98 |
| $r$ (m) | 0.76 | 0.78 | 0.80 | 0.82 | 0.84 | 0.86 | 0.88 | 0.90 | 0.92 | 0.94 | 0.96 | 0.98 | 1.00 | |
| $c$ (m) | 0.120 | 0.112 | 0.104 | 0.095 | 0.086 | 0.076 | 0.067 | 0.057 | 0.047 | 0.038 | 0.028 | 0.019 | 0.010 | |
| $\beta°$ | 26.40 | 25.85 | 25.33 | 24.85 | 24.40 | 24.01 | 23.67 | 23.41 | 23.28 | 23.39 | 23.99 | 25.81 | 32.23 | |

*Code availability.* DUST is open source and available online at https://www.dust.polimi.it/. The input files for running the DUST simulations

are available at https://github.com/polimi-saslab/pyDUST_windplane.

*Author contributions.* FT conceptualized and developed the research methods, produced the results and wrote the draft version of the paper. MG contributed to the conceptualization of the paper, to the development of the aerodynamic model and revised the paper. GC contributed to the conceptualization of the paper, to setting up the DUST simulations, and revised the paper. LF contributed to the conceptualization of the paper, to the optimal problem formulation, and revised the paper.

*Competing interests.* The authors declare no competing interests

*Acknowledgements.* This research has been supported by Fondazione Cariplo under grant n. 2022-2005, project "NextWind - Advanced control solutions for large scale Airborne Wind Energy Systems", by the Italian Ministry of University and Research under grants "DeepAirborne - Advanced Modeling, Control and Design Optimization Methods for Deep Offshore Airborne Wind Energy" (NextGenerationEU fund, project P2022927H7), the Extended Partnership "NEST - Network 4 Energy Sustainable Transition", by the MERIDIONAL project,

which receives funding from the European Union's Horizon Europe Programme under the grant agreement No. 101084216, and by "IEA Wind Task: Airborne Wind Energy" funded by the Danish EUDP, Energy Technology Development and Demonstration Programme, J.134-21007.



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
