# Peer review of "Concurrent aerodynamic design of the wing and the turbines of airborne wind energy systems"

_Wind Energy Science, 2025_

## Author Comment (AC1)

Dear Editor, dear Reviewers,

Thank you very much for your comments and for taking the time to review our work.

In the following we go through your *comments* and provide, for each one, both our *responses* and the *actions* we have taken to accommodate your feedback in the revised manuscript.

When answering your comments we refer to the *figures and lines number* of the manuscript with the highlighted changes.

Best regards,

The Authors

**Reply to Anonymous Referee #1**

[https://doi.org/10.5194/wes-2025-134-RC1]

**Review Comment**

**GENERAL COMMENTS**

I find this to be an overall well-done paper filling a gap in the current literature. While analogous studies have been performed for propulsive cases, this paper provides somewhat detailed findings for wind turbinewing interactions currently missing from the literature. The methods and models are generally clear and the results appear to be reasonable. I also appreciate the recommendations throughout this work concerning potential future work stemming from the studies presented herein.

**Authors Answer & Actions**

Thank you very much for the detailed review and the many suggestions. We feel that the paper increased in clarity thanks to your feedback.

**Review Comment**

SPECIFIC COMMENTS/QUESTIONS

**Overview:**

- 1. I think a more explicit highlighting of what your paper adds would greatly improve the impact. I see several times throughout that your results "confirm the results of..." such and such author, but it's not quite explicit how your work adds to the current literature. I would recommend making this clearer in the introduction as well as the conclusion, then include some more high impact statements in the discussions of your studies throughout to make clear what the new results are and why they are important.
- 2. I would like to see more discussion in conjunction with
  - o Figures 10-13
  - Analysis out of the design point section

**Authors Answer & Actions**

1. We highlighted the novelty by adding "novel" or "new" when talking about the methodology and the results in the abstract and in the conclusions. The model is new and the conclusions are almost all new: we comment with "confirm the results of..." when the cited papers take general conclusions which are valid also for this case. We fell that the impact statements in the conclusions are enough to show the new contributions of this paper.

**2. We modified as follows:**

- We restructured the whole section by removing figure 11 and adding two more figures (12 and 13). We added more explanations on the CL and AR trends (Also the second reviewer asked for more details here). The new figures are given to give a more incremental understanding of the results. This part has been restructured for improving the clarity.
- We added some considerations in the "Analysis away from the design point" section. We added a consideration to the fact that the design point is far from stall and we explained better the control considerations.

**Review Comment**

**Abstract:**

- 1. At this point, I'm looking forward to seeing more explanation about how reducing induced drag of the wing increases power production of the turbines. This doesn't seem like a direct consequence, and could maybe use some minor elaboration at this point.
  - a. After one read through, I don't think this was satisfactorily answered, or perhaps not pointed out explicitly.
- 2. "Especially when they are placed near at the wing tips" Does the wake swirl reduce induced drag when the turbines are not placed at the wing-tips? It seems like you'd need some destructive interaction with the wing tip vortices, so you'd have to at least be close.
- 3. "Moreover, conventional efficient airfoils are found to be optimal for windplanes." such as? And compared to what? Are NACA4421 airfoils these conventional efficient airfoils? If not, why use them instead of the optimal ones for your studies?
- 4. What do you mean by an optimal trapezoidal wing with constant twist? Optimal in what way? I'm not seeing how you could get an elliptic lift distribution with that planform? Also, optimal wing only, or optimal with the interactional aerodynamics of the turbines?
- 5. I'd like to see a quick summary of the major results from the DUST studies you did. What did those studies lead to? What are the major takeaways?

**Authors Answer & actions**

- 1. It is not a direct consequence indeed. Reducing the induced drag means increasing the flight speed (i.e. the wing speed ratio lambda), which leads to an increase in power production. We added this consideration in the abstract and added Fig. 12 to show this.
- 2. Yes, the induced drag decreases even if they are not added at the wingtips, leading to an increase in power (see fig 17). However, the largest increase in power production is achieved when they are placed at the wingtips. We removed this sentence, as in the abstract is creating confusion as you pointed out.
- 3. We added this consideration as unconventional airfoils, which maximize the metric Cl^2/Cd^3, are typically used in Airborne Wind Energy. By reformulating the problem to maximize the power coefficient used in this paper, conventional efficient airfoils maximizing Cl/Cd are optimal. NACA4421 are efficient airfoils. We rephrase the sentence in the abstract and added a detailed explanation of this point in the introduction (from line 48) and in sect 3.1 (the paragraph staring at line 343).
- 4. We refer to "optimal" as the wing which maximizes power production within the modelling assumptions we set. So, it is not necessarily with an elliptical lift distribution. We modify this sentence in the abstract accordingly. In the manuscript, following the optimization problem formulation, it should be clear that the optimal wing is the constant twist trapezoidal wing which maximizes power production.
- 5. We added a sentence stating that the maximum in power production is found close to the design point and explained in more detail in the text and conclusions. For Airborne wind energy systems designs in the literature the maximum power is found at the maximum lift coefficient, which is close to stall.

**Introduction**

- 1. "at very high lift coefficients" how high is very high? 2? 5?
- 2. How do you know that KiteKraft designs their turbines and wings separately? Do you have a source for that?
- 3. Same comment about conventional airfoils as in the abstract.
- 4. Three turbine positions are considered: in front, at the wing tips and above or below the wing" a figure would be immensely helpful here, it seems like all these are in front. Is that a position? Or are wing tips and above and below the three positions? Why can't the turbines be in front, at the wing tips and also above and below simultaneously? I suppose I'd like to see some clarification on what is meant by "in front."
  - I see you have figure 6 that explains this well. I think some re-wording here could be sufficient instead of another figure.

**Authors Answer & Actions**

- 1. From the cited paper, CL=4.5. We added the information
- 2. Yes, the source is given in the sentence before. We reformulated the two sentences to make this understandable.
- 3. We added a detailed clarification.
- 4. We give some more information here, but later in the text, as you noted, we give more details.

**Review Comment**

Windplane steady-state model

• I'm unsure what is meant by: "defined to inform about the aerodynamic force applied to the wind"

**Authors Answer & Actions**

It is the aerodynamic force which does work to the wind (i.e. the component of the aerodynamic force which is parallel to the wind direction and thus slows it down). We clarified it.

**Review Comment**

Onboard turbines model

• "The importance of this effect increases as the tip speed ratio is decreased." — Please expound.

**Authors Answer & Actions**

At low tip-speed ratio the wake rotation increases. Capturing this effect is very important in studies such this one because they wake rotation is responsible for the change in inflow angle at the wing. We specified it in the text.

**Review Comment**

Windplane aerodynamic design problem

- 1. For the pylon mounted turbines, how far above the wing did you place the rotors to justify turning off the interactional aerodynamics?
- 2. I would like to see more discussion on figures 10-12 if you're going to include them. Same with figure 13 (you also have just a single sentence paragraph).
- 3. You may want to quantify in some way why a fixed 1 meter radius is reasonable (line 365).
- 4. I am not following how your study makes conclusions about conventional efficient airfoils. It seemed that your airfoils were somewhat notional and not directly tied to some sort of convention.
- 5. Please provide some quantification (based on your study) of how much power can increase with turbines placed at wing tips. (How much is "considerable"?)

**Authors Answer & Actions**

- 1. That's a challenging question; we would suggest at least half rotor radius from the wing. This topic is investigated in depth in the paper "Mehr, J. et al: Interactional Aerodynamics Analysis of a Multirotor Energy Kite, Wind Energy, 2024". Case c) in our sensitivity analysis wants to look at the extreme idealized case in which there is no interaction, rather than investigating a realistic case in which there is some interaction. We add a sentence pointing to this paper in the case description.
- 2. We removed the figure showing the CPt/CTt as it was somehow redundant with the last figure of the section (Fig 20); we added some comments for the figures and added 2 extra figures to better explain the results. The new figure 12 shows the ratio between CL and CD. The second new figure 13 shows the ratio between the turbines' thrust and the total drag. For idealized turbines, this ratio should be 0.5 (Loyd 1980).
- 3. We added a comment. The rotor radius in a real system will be determined by a trade-off between other disciplines influencing the design. We feel that Rt=1 m is a meaningful first guess, as the curve CP-Rt is approaching the plateau.
- 4. We added a paragraph about this right after the description of Figure 8. We hope that now the conclusions about the optimal airfoils are clearer.
- 5. We quantified it (approx. 5%)

Optimal aerodynamic design

• Figure 19 — what causes the blade twist to rise near the tip?

**Authors Answer & Actions**

The reason is the modification that we carry out at the root and at the tip, which are explained at the end of Section 3.0 "The turbine design features a constant lift coefficient along the blade span at the design tip speed ratio. In order to achieve a realistic design, the design lift coefficient at the blade tip is lowered to delay stall at lower tip-speed ratios and the chord at the blade root is increased to allow for the hub connection. For these reasons, the chord at the root and at the tip is slightly widened, while the twist is adjusted to keep the same lift force."

We add a sentence explaining it when introducing figure 22.

**Review Comment**

Isolated turbine

- 1. It sounds like you ran 8 revolutions based on Niro's work, is that long enough to achieve steady state?
- 2. Line 415 please quantify "really close"

**Authors Answer & Actions**

- 1. We added a convergence study in appendix A.
- 2. The values of the CPt and CTt obtained with the two models are given in the same sentence, we consider this as a quantification.

**Review Comment**

Analysis out of the design point

- 1. I'm not sure I'd say "exclusively influenced," it seems like you're just trying to say that you perform sweeps of wing angle of attack and rotor tip speed ratio. Unless you've decoupled the interactional aerodynamics, then the effects are not exclusive.
  - I take it back, but you do say this applies to the whole study and then immediately show figures with and without interactional aerodynamics, so something still needs to change to make this clearer.
- 2. "To reduce power production," why would you want to reduce power production? It seems like the sentence is backwards. I see why you'd need to reduce loads and the result would be reduced power, but it sounds here like reducing power is your goal, and reduced loads is a result.

- This sentence in general (line 477-478) is confusing. I'd probably make it a couple sentences or more and be a bit more clear what you're trying to get across.
  - Carefully examining figures 32 and 33, I think I'm following, but it should be clearer in the text that you're trying to say that reducing angle of attack is a more effective way to reduce loads without losing too much power since the power coefficient is much more sensitive to tip speed ratio that angle of attack.
- 3. I'd like to see a bit more discussion of the results in this section. You have some good takeaways already, but it still feels a bit light relative to how much information should be able to be communicated with this amount of figures.

**Authors Answer & Actions**

- 1. We added "isolated" wing and "isolated" turbines to clarify this point.
- 2. We reformulated the sentence in "To reduce the aerodynamic power after the rated wind speed" to make clear that the case in which we want to reduce power is after rated power is reached. We restructured the sentence in more sentences, to explain better this concept.
- 3. We added a comment on the fact that the optimum is found far from stall.

**Review Comment**

**Appendix A**

• The comment here on at least 6 revolutions should probably be included in the main text where you also cite Niro 2024.

**Authors Answer & Actions**

We added the conference study and removed the reference to Niro 2024.

**Review Comment**

**SUGGESTED TECHNICAL CORRECTIONS**

**Overview:**

- 1. Miscellaneous Grammar:
  - I would prefer the Oxford comma to be used in lists throughout. It is present in some cases, but only a few.
    - If not including it, remove it everywhere for consistency.
  - There are several end of sentence parentheticals, especially in the abstract/intro that reduce the flow of the text. Some minor sentence restructuring would be in order for easier readability.
  - There are quite a few pluralization and possessive errors throughout (missing apostrophes for possessives, plurals where they shouldn't be, etc.). I've tried to highlight the ones I noticed, but another read through would be good to fix those.

**2. Figures**

- I find the tikz-type figure coloring to be somewhat haphazard. I feel like there are quite a few opportunities to standardize colors to mentally link between figures as well as make things more readable.
- Although not necessary, I generally recommend line plots (figure 7 to the end), not have grids, upper and right spines (except for multi-axis plots), and have no border around legends.
  - These items are generally not required to understand the messages being communicated by the plots and removing them reduces extraneous information making it easier for readers to digest the important contents.
- Similarly, rotating the y-axis labels 90 degrees makes it just a bit easier to read and there is typically sufficient space to allow for a wider y-axis label.

**Authors Answer & Actions**

- 1. Miscellaneous Grammar:
  - We included it everywhere for consistency
  - We went through them and rephrased when possible.
  - We went through them and correct them. There is a final English proofreading from the journal prior to the publication.
- 2. Figures: we prefer to leave plots as they are, as the suggestions are a matter of aesthetics.

**Abstract:**

- 1. Is "windplane" plural? AWE systems is plural, and then you say "their power production" (also plural). I'd expect it to be windplanes.
- 2. This first sentence is a bit long with a parenthetical and a second clause. At the very least, put a semi-colon instead of a comma before "but," however I would recommend splitting it into multiple sentences and be more clear on what the largely unexplored problem is specifically (what does "it" refer to? The aerodynamic design, the determination of power production?) Alternatively, rearranging the sentence could make it shorter and clearer as well.
- 3. I'm a fan of using the oxford comma rather than leaving it out in sentence 2 (between wing and "and their interactional"). You use it later in a similar sentence, so either way, just be consistent.
- 4. More comma inconsistency in line 11 starting with "the results from the vortex models..." In this case I'd definitely recommend a comma after "vortex lattice method".
- 5. You may consider reordering this sentence as well to not have "finding very good agreement" tacked on the end but rather closer to the beginning of the sentence.

**Authors Answer & Actions**

- 1. We refer to windplane as the aircraft of fly-gen AWES, so it is singular.
- 2. We rephrase the sentence for clarity.
- 3. We added the Oxford comma.
- 4. We added the Oxford comma.
- 5. We rephrase it.

**Review Comment**

**Introduction**

- 1. "WESs can be classified in crosswind" classified as crosswind...
- 2. "crosswind, tether aligned and rotational" aligned, **or** rotational... (unless they can be classified as more than one, then and/or, but either way, add that comma in)
- 3. "roughly perpendicular to the wind direction and generate" I would add **either** before "generate" here. I almost commented that you forgot pumping AWES since the beginning of the sentence didn't indicate that there was a comparison to be made.
- 4. Comma after Windlift (2025)
- 5. Line 30: "multi-elements airfoils" —> muli-element airfoils.
- 6. "onboard turbines" —> onboard turbine
- 7. I'm seeing a lot of "after thought" type phrases at the end of sentences that make it harder to follow than it needs to be. Instead of tacking on parentheticals at the end of sentences, it may be better to generally restructure things to be more direct in the first place rather than requiring constant mental backtracking to put all the pieces together.
- 8. Either need to say **The** Windlift design, or Windlifts design.
- 9. Comma needed after disclosed

**Authors Answer & Actions**

- 1. Modified accordingly.
- 2. Modified accordingly.

- 3. Modified accordingly.
- 4. Modified accordingly.
- 5. Modified accordingly.
- 6. We prefer to leave the plural here.
- 7. We tried to rephrase some sentences accordingly.
- 8. Modified accordingly.
- 9. Modified accordingly.

**Windplane steady-state model**

- 1. Because the subscript is small, the tau and r subscripts for the coordinate system are somewhat difficult to differentiate. It might be nice to use a different symbol for the tangential direction, perhaps theta, for increased clarity.
- 2. Figure 1 you have v\_w and Phi in the figure but don't mention them in the caption (and I don't see v\_w defined nearby). If they aren't important enough to mention, then maybe they don't need to be in the figure in the first place? If they are vital, then they should be mentioned.
- 3. R\_0 is given in the nomenclature, but not in conjunction with equation 1 like the other variables are. It would be nice to have it with everything else and perhaps included in figure 1 or 2 somehow.
- 4. Figure 2 Is there any significance to the various colors used? I notice v\_w matches colors between figure 1 and 2, but Phi does not. I almost thought there was color coordinates for the expressions in equation 1, but it doesn't appear to be so. Also, the red, magenta, brown, and orange are difficult to differentiate without careful (and very zoomed in) consideration. I would recommend being a bit more judicious with color usage, and if so many colors are needed, I would recommend using more contrasting colors for terms in near proximity (for example, the v\_w, u, v\_a triangle is nicely contrasted, but does it require so many colors?)
  - The xi\_tb/2 term could afford to be a bit closer to the arrow it's associated with (you could also move the arrow direction since it's just radius)
  - There's extra space behind "wake" in the wing wake box, maybe center "wing wake" in the box?
  - May want to include m with u^2/R\_0 in that box just for consistency so there's no confusion that those are the same term
- 5. It is exceptionally inconvenient to refer to figure 5 seven pages before you show it. I would either remove this reference on page 5, or if it's vital for understanding at this point, move it up.
- 6. The in-line equations starting on page 6 (starting at equation 11, but used throughout the paper) are somewhat difficult to read. If it's not a required style, I would break those out into multi-line expressions to increase readability and the reader's ability to follow the math.

**Authors Answer & Actions**

- 1. We modified it accordingly.
- 2. We added them in the caption.
- 3. We added R 0 to the list.
- 4. We tried to fix the figures following all your comments.
- 5. We removed the reference to Fig. 5
- 6. This will be fixed in the final version of the paper, which will be of two columns.

**Review Comment**

**Onboard turbines model**

- 1. "In order to evaluate the onboard turbines performance..." should put an apostrophe after the s in turbines (**turbines**') to indicate possession for multiple turbines.
- 2. "effect of their wake" —> effect of their wakes

3. I'd recommend not using the BEM acronym without the full term, especially since this is the only place you use it, so you don't need the acronym anyway.

**Authors Answer & Actions**

Modified accordingly.

**Review Comment**

**Wing model**

- 1. Seems like you'd want figure 3 at the top of page 9, right when you reference it.
- 2. Figure 3 you could use far fewer vortex elements so that your labels aren't so crowded.
  - Back in figure 2, you used dotted lines at the end of the tether wake to indicate they extended back further. You specifically mention that the trailing vortices extend to the end of the fluid domain, so it would be a nice touch to add similar dots to those trailing vortex lines.
  - Also, if you lighten the shading on the wing, it would make things every so slightly easier to
- 3. Figure 4 it would be nice if the arrow heads were a bit bigger, especially since several of them overlap with other lines or X's.
  - I like that you have an X at the 3c/4 point that ties figure 3 and 4 together, but you may want to match a symbol between figure 3 and 4 for the 1c/4 point to further improve clarity (maybe a blue circle?)

**Authors Answer & Actions**

- 1. This will be fixed with the final version
- 2. We modified accordingly
- 3. We modified accordingly

**Review Comment**

Windplane aerodynamic design problem

- 1. Instead of saying "following figures" when talking about dashed lines, why not just indicate the range of figure numbers?
  - o I also don't think you need to include m-dashes here. Everyone knows what a dashed line is.
  - Same comments on dotted lines.
- 2. Figure 6 the figure makes it look like the rotors are placed at different streamwise positions between each case (6a makes it look like the rotors are nearly touching the leading edge, 6b shows them out front, and 6c is in between), is that intentional?
- 3. Line 334-335 "reducing the induced drag in the wing aerodynamics" —> either "in" should either be of or on depending on what you're trying to say.
- 4. "Designs with lower tip speed ratios are typically preferable to reduce the noise emission and the erosion." citation?

**Authors Answer & Actions**

- 1. We modified it accordingly.
- 2. No, it was not intentional. We modified the figure accordingly.
- 3. Modified with "of"
- 4. We removed the mention to the erosion because we did not find any literature linking the erosion of air-propellers to erosion. These considerations are coming from our knowledge in wind energy, where there are many studies finding correlation between the erosion and the rotational speed during precipitation. We gave a reference for the noise emission for air-propellers.

**Review Comment**

- 1. Line 394: "increase" -> increases
- 2. "This low value of  $\lambda t$  is also good to limit the blade erosion and the acoustic emission." again, citation?
- 3. "Modeling the wing with the present formulation, which is accurate even for low aspect ratio, the spanwise efficiency of the wing, accounting for the onboard turbine's influence, can take values above one." multiple parentheticals make this sentence difficult to parse.

**Authors Answer & Actions**

- 1. Modified accordingly
- 2. Modified accordingly
- 3. Rephrased.

**Review Comment**

Isolated turbine

- Line 411-412: "turbines blades" —> turbines' blades
- Comma after Appendix A on line 412

**Authors Answer & Actions**

Modified accordingly

**Review Comment**

Isolated wing

• Figures 26 and 27, I don't think you need a dash to indicate that Cl and Cd are unitless. You haven't used that for any plots up to this point for other non-dimensional numbers.

**Authors Answer & Actions**

Modified accordingly

**Review Comment**

Interactional aerodynamics

- 1. You could probably make figures 28 and 29 a single figure with 2 sub-figures for more easy viewing.
- 2. I don't think figure 30 is necessary to include. It doesn't appear to add anything to the discussion.
  - If you do keep it, I'd suggest moving it up to where you describe the discretization used in DUST.
  - You can also make some adjustments in Paraview (such as adding some opacity and maybe gaussian blur to the particles) to make it a bit more aesthetic and de-emphasize the particles if you want to highlight the discretization.

**Authors Answer & Actions**

- 1. Since we modified these two figures to add the isolated wing solution, we prefer to leave them separated.
- 2. We moved the figure to the section where we talk about the parametrization in DUST and adjusted the colors in Paraview

**Review Comment**

Analysis out of the design point

- 1. Perhaps analysis **away from** (or **off of**) the design point? Typically I've seen this called "off design cases"
- 2. "Aerodynamics is exclusively" —> aerodynamics are exclusively (shows up multiple times

3. Figures 31and 32 — without those tiny numbers, I'd not be able to tell the difference between these figures. I feel like a line plot at slices of lambda and alpha through the maximum Cp operating points would be clearer to compare the power with and without interactional effects.

**Authors Answer & Actions**

- 1. We modified to "off-design" in the title.
- 2. Modified accordingly.
- 3. We would prefer to leave the two-dimensional plot. To make it clearer, we change the background, the line and the text size.

**Review Comment**

**Conclusions**

• A few spelling errors and missing apostrophes.

**Authors Answer & Actions**

We checked them and modified.

**Review Comment**

**Nomenclature**

• You may consider using lower-case "c" for the section drag coefficients. You already use lower-case subscripts, but it could potentially be easier to differentiate if the C's were also lower-case.

**Authors Answer & Actions**

We prefer to leave it like this, as it appears to be aligned with literature.

**Reply to Anonymous Referee #2**

[https://doi.org/10.5194/wes-2025-134-RC2]

**Review Comment**

General comments

Engineering models for turbine and wing aerodynamics are used to optimize a power coefficient and then validated against a higher fidelity aerodynamic analysis. The model itself is highly simplified, but presents a method that can be used to optimize the aerodynamics of an AWE system, rather than designing one aerodynamic component at a time.

**Authors Answer & Actions**

Thanks you very much for taking the time to review our paper. We feel that the paper has improved in quality thanks to your review.

**Review Comment**

Specific comments

25

The article mentions the different flight modes required for a flygen system and other flygen AWE companies. The Kitekraft and the Makani M600 designs use 8 rotors and the Windlift and Makani Wing 7 designs use 4 rotors which allow for pitch, roll, and yaw control using the rotors in takeoff and landing. The model in this paper uses two rotors. It's worth mentioning how the design is intended to perform those flight modes, especially because of the emphasis on the benefits of putting rotors at the wingtips.

60

The article states that the rotors are designed for the generation phase only, not considering whether they generate enough thrust in hover. It's worth looking at how stalled the rotors are in static thrust, and whether the analysis tools are reliable for that operating point.

**Authors Answer & Actions**

These points are interesting, however we feel that describing how these systems could take-off and land would be out of scope. We indeed put a lot of emphasis on the benefits of placing the rotors at the wingtips to show the potential and to develop the methodology which could be later applied to more realistic designs. Later designs might have more than two rotors or other engineering solutions to control pitch, roll, and yaw in these phases. We add a comment to the fact that this framework is also applicable to designs with more than 2 turbines after Eq 10, where the number of turbines is first introduced.

In the research group of the first and last author, we are focusing on the design and control of the VTOL phase from a theoretical and experimental point of view. As we will soon submit works related to the VTOL phase, we prefer to leave the solutions to the problems you raised out from this paper.

**Review Comment**

**Authors Answer & Actions**

330

"Higher taper ratios are preferable if the turbines are placed at the wing tips. This is because higher taper ratio wings have more lifting area behind the turbines, enhancing their beneficial effect on the power production."

I would guess that higher taper ratios are preferable because the turbines at the wing tips are reducing airspeed and lift, so a higher taper ratio produces a distribution closer to eliptical. I'd be interested to see if this method considers the most eliptical lift distribution to be the most optimal or whether there are other factors.

**Authors Answer & Actions**

That's an interesting comment. However, the turbines reduce lift of a small quantity, or they even increase it. The effects of the turbines is two-fold: decreasing the dynamic pressure of the turbines' axial induction  $a_t$ :  $\frac{1}{2}\rho(1-a_t)^2u^2$ , where  $a_t$  is shown in Fig. 23, and increasing the lift coefficient of  $\Delta C_l = C_{l,a}\alpha_{i,t}$ , where  $\alpha_{i,t}$  is the induced change in angle of attack at the wing due to the turbine (Fig. 24) and  $C_{l,a}\approx 2\pi$ . Inserting the values from the figures and neglecting that the bound circulation with and without turbines is different, we find that the lift does slightly increase.

Taking  $\alpha_{l,t}=3^{\circ}$  ,  $a_{t}=0.05$  and  $\textit{C}_{l.a}=5.5$  1/rad

$$\frac{l_{interactional}-l_{isolated}}{l_{isolated}} = \frac{\frac{1}{2}\rho c \left(c_l + C_{l,a}\alpha_{i,t}\right)(1-a_t)^2 u^2 - \frac{1}{2}\rho c c_{lu^2}}{\frac{1}{2}\rho c c_{lu^2}} \approx C_{l,a}\alpha_{i,t}(1-a_t)^2 \approx 0.25$$

So, the lift with the turbines increases of approx. 25%. In practice, this increase is not reached because the bound circulation in the sections behind the turbines increases and thus the induced velocities to increase, the angle of attack to decrease and the lift to decrease. To show this, we modified Fig 31 and 32 to include also the trends of the isolated wing (with DUST), thus showing the effect of the turbines. As you can see now, the lift stays almost unchanged with the turbines. We commented in the text consequently (line 514), but we prefer to not write this last equation because it is derived by neglecting the change in bound circulation, and it could be misinterpreted. We leave the comment about the figure of the taper ratio (15) unchanged as we think it is correct.

Elliptical lift distribution are optimal in minimizing the induced drag because they generate constant trailed vorticity and thus they induce constant velocities along the wing span. This is true for high aspect ratio wing, where the effect of the 2D bound vorticity is negligible (Eq. 33), and the Weissinger lifting line collapses into the Prandl lifting line. In our case, the optimal aspect ratio turns out to be low (approx. 5), and the Prandl lifting line does not match the vortex lattice results. This was the reason for developing a Weissinger lifting line to model the wing (collocations points at ¾ chord). With this method, the optimal lift distribution is not necessarily an elliptical distribution because of the nonlinear effects due to the 2D bound vorticity. We remarked this at line 452.

In our paper, we approach the problem by looking at a simple design with trapezoidal planform and constant twist. The choice of this simple wing is to reduce the design space to a few main parameters and get to a baseline design that can be (largely) improved later. This wing parametrization prevents the optimizer to find the "real" aerodynamic optimum, for which we need to add more design variables to the problem. We added a sentence after the problem statement to highlight the need of understanding in more details the optimal lift distribution (sect 3.0)

By analyzing figures 29, we can understand some qualities of the induced velocities along the isolated wing. The lift coefficient distribution shows us that the angles of attack along the wing changes, meaning that also the induced velocities change (recall that the twist angle is constant). This is already pointing out that this wing does not have constant induced velocities along the wing. We added a comment on this at line 501.

**Review Comment**

320

"The trends show that it is optimal to design the wing such that it operates the airfoils at their maximum efficiency."

Because most drag in an airborne wind system is induced drag and tether drag rather than wing airfoil profile drag, the highest system lift to drag ratio shouldn't usually occur at the same angle of attack as the airfoil max lift to drag ratio. I'd like to see an explanation on why figure 7 looks like it does (what are the induced drag, tether drag, rotor force, and profile drag doing around those points)?

**Authors Answer & Actions**

The systems analyzed and developed in the literature are designed based on the power harvesting factor  $\xi_P$ , which uses the wing planform area  $A=\frac{b^2}{AR}$  as the reference area. The power harvesting factor can be written as the power coefficient times the wing aspect ratio

$$\xi_P = \frac{P}{\frac{1}{2}\rho\pi \frac{b^2}{AR}v_w^3} = \pi C_P AR$$

This objective leads to designs with high aspect ratio and high design lift coefficients (the highest possible, typically above 2), with dedicated airfoils designed for this task. These airfoils are designed to maximize the metric Cl^3/Cd^2 at the design lift coefficient. When analyzing these systems, the conclusion you make is correct.

We perform this sensitivity analysis to show how this new power coefficient changes the system design. In this case, it is optimum to design lower aspect ratio wings and operated them at the point of airfoil max CI/Cd. In this case, the induced drag is still the dominant drag component (approx. 40 %), followed by the turbines' thrust (approx. 35%), the tether drag (approx. 15%) and profile drag (approx. 10%). Note that the optimal turbines' thrust is approx. half of the total drag, as found by Loyd (figure 13)

To explain better these trends, we added a Figure 12 and 13 and we added a paragraph explaining how the two objective functions leads to different system designs in the introduction and at line 343.

We give the contributions of each drag term are given for the case study in Table 2 and in the text introducing the table.

**Review Comment**

**Technical corrections**

50

"they reduce the apparent wind speed felt by the involved wing aerodynamic sections, which in principle would lead to a lower lift force. On the contrary, airplanes with propellers in front of the wing achieve an increase in aerodynamic lift."

Awkward language sounds like its arguing the turbines might increase lift. I'd change it to "they reduce the apparent wind speed felt by the involved wing aerodynamic sections, which reduces lift force. Conversely, airplanes with propellers in front of the wing achieve an increase in aerodynamic lift"

**Authors Answer & Actions**

We modified the sentence according to your suggestion.